# Dangers of Bayesian Model Averaging under Covariate Shift

**Pavel Izmailov**          **Patrick Nicholson**          **Sanae Lotfi**          **Andrew Gordon Wilson**
NYU          Covera Health          NYU          NYU

## Abstract

Approximate Bayesian inference for neural networks is considered a robust alternative to standard training, often providing good performance on out-of-distribution data. However, Bayesian neural networks (BNNs) with high-fidelity approximate inference via full-batch Hamiltonian Monte Carlo achieve poor generalization under covariate shift, even underperforming classical estimation. We explain this surprising result, showing how a Bayesian model average can in fact be problematic under covariate shift, particularly in cases where linear dependencies in the input features cause a lack of posterior contraction. We additionally show why the same issue does not affect many approximate inference procedures, or classical maximum a-posteriori (MAP) training. Finally, we propose novel priors that improve the robustness of BNNs to many sources of covariate shift.

## 1   Introduction

The predictive distributions of deep neural networks are often deployed in critical applications such as medical diagnosis [Gulshan et al., 2016, Esteva et al., 2017, Filos et al., 2019], and autonomous driving [Bojarski et al., 2016, Al-Shedivat et al., 2017, Michelmore et al., 2020]. These applications typically involve *covariate shift*, where the target data distribution is different from the distribution used for training [Hendrycks and Dietterich, 2019, Arjovsky, 2021]. Accurately reflecting uncertainty is crucial for robustness to these shifts [Ovadia et al., 2019, Roy et al., 2021]. Since Bayesian methods provide a principled approach to representing model (epistemic) uncertainty, they are commonly benchmarked on out-of-distribution (OOD) generalization tasks [e.g., Kendall and Gal, 2017, Ovadia et al., 2019, Chang et al., 2019, Dusenberry et al., 2020, Wilson and Izmailov, 2020].

However, Izmailov et al. [2021] recently showed that Bayesian neural networks (BNNs) with high fidelity inference through Hamiltonian Monte Carlo (HMC) provide shockingly poor OOD generalization performance, despite the popularity and success of approximate Bayesian inference in this setting [Gal and Ghahramani, 2016, Lakshminarayanan et al., 2017, Ovadia et al., 2019, Maddox et al., 2019, Wilson and Izmailov, 2020, Dusenberry et al., 2020, Benton et al., 2021].

In this paper, we seek to understand, further demonstrate, and help remedy this concerning behaviour. We show that Bayesian neural networks perform poorly for different types of covariate shift, namely test data corruption, domain shift, and spurious correlations. In Figure 1(a) we see that a ResNet-20 BNN approximated with HMC underperforms a maximum a-posteriori (MAP) solution by $25\%$ on the *pixelate*-corrupted CIFAR-10 test set. This result is particularly surprising given that on the in-distribution test data, the BNN outperforms the MAP solution by over $5\%$.

Intuitively, we find that Bayesian model averaging (BMA) can be problematic under covariate shift as follows. Due to linear dependencies in the features (inputs) of the training data distribution, model parameters corresponding to these dependencies do not affect the predictions on the training data. As an illustrative special case of this general setting, consider MNIST digits, which always have black corner pixels (*dead pixels*, with intensity zero). The corresponding first layer weights are always multiplied by zero and have no effect on the likelihood. Consequently, these weights are simply

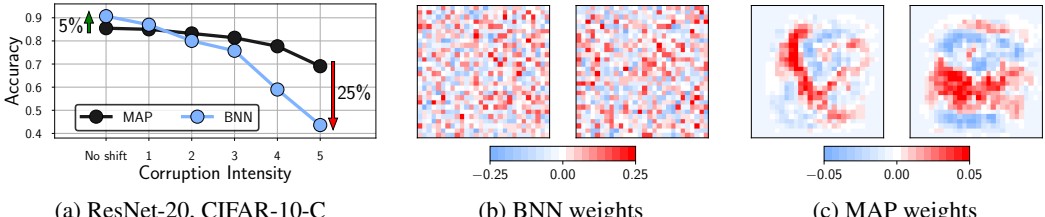

| (a) ResNet-20, CIFAR-10-C | (b) BNN weights | (c) MAP weights |

Figure 1: **Bayesian neural networks under covariate shift**. **(a)**: Performance of a ResNet-20 on the *pixelate* corruption in CIFAR-10-C. For the highest degree of corruption, a Bayesian model average underperforms a MAP solution by 25% (44% against 69%) accuracy. See Izmailov et al. [2021] for details. **(b)**: Visualization of the weights in the first layer of a Bayesian fully-connected network on MNIST sampled via HMC. **(c)**: The corresponding MAP weights. We visualize the weights connecting the input pixels to a neuron in the hidden layer as a $28 \times 28$ image, where each weight is shown in the location of the input pixel it interacts with.

sampled from the prior. If at test time the corner pixels are not black, e.g., due to corruption, these pixel values will be multiplied by random weights sampled from the prior, and propagated to the next layer, significantly degrading performance. On the other hand, classical MAP training drives the unrestricted parameters towards zero due to regularization from the prior that penalizes the parameter norm, and will not be similarly affected by noise at test time. Here we see a major difference in robustness between optimizing a posterior for MAP training in comparison to a posterior weighted model average.

As a motivating example, in Figure 1(b, c) we visualize the weights in the first layer of a fully-connected network for a sample from the BNN posterior and the MAP solution on the MNIST dataset. The MAP solution weights are highly structured, while the BNN sample appears extremely noisy, similar to a draw from the Gaussian prior. In particular the weights corresponding to dead pixels (i.e. pixel positions that are black for all the MNIST images) near the boundary of the input image are set near zero (shown in white) by the MAP solution, but sampled randomly by the BNN. If at test time the data is corrupted, e.g. by Gaussian noise, and the pixels near the boundary of the image are activated, the MAP solution will ignore these pixels, while the predictions of the BNN will be significantly affected.

Dead pixels are a special case of our more general findings: we show that the dramatic lack of robustness for Bayesian neural networks is fundamentally caused by *any* linear dependencies in the data, combined with models that are non-linear in their parameters. Indeed, we consider a wide range of covariate shifts, including domain shifts. These robustness issues have the potential to impact virtually *every* real-world application of Bayesian neural networks, since train and test rarely come from exactly the same distribution.

Based on our understanding, we introduce a novel prior that assigns a low variance to the weights in the first layer corresponding to directions orthogonal to the data manifold, leading to improved generalization under covariate shift. We additionally study the effect of non-zero mean corruptions and accordingly propose a second prior that constrains the sum of the weights, resulting in further improvements in OOD generalization for Bayesian neural networks.

Our code is available here.

## 2  Background

**Bayesian neural networks.**    A Bayesian neural network model is specified by the prior distribution $p(w)$ over the weights $w$ of the model, and the likelihood function $p(y|x, w)$, where $x$ represents the input features and $y$ represents the target value. Following Bayes' rule, the *posterior* distribution over the parameters $w$ after observing the dataset $\mathcal{D} = \{(x_i, y_i) | i = 1, \ldots, n\}$ is given by

$$p(w|\mathcal{D}) = \frac{p(D|w) \cdot p(w)}{\int_{w'} p(D|w') \cdot p(w') dw'} = \frac{\prod_{i=1}^{n} p(y_i|x_i, w) \cdot p(w)}{\int_{w'} \prod_{i=1}^{n} p(y_i|x_i, w') \cdot p(w') dw'}, \tag{1}$$

where we assume that the likelihood is independent over the data points. The posterior in Equation 1 is then used to make predictions for a new input $x$ according to the *Bayesian model average*:

$$p(y|x) = \int p(y|x, w) \cdot p(w|\mathcal{D})dw. \qquad (2)$$

Unfortunately, computing the BMA in Equation 2 is intractable for Bayesian neural networks. Hence, a number of approximate inference methods have been developed. In this work, we use full-batch HMC [Neal et al., 2011, Izmailov et al., 2021], as it provides a high-accuracy posterior approximation. For a more detailed discussion of Bayesian deep learning see Wilson and Izmailov [2020].

**Maximum a-posteriori (MAP) estimation.** In contrast with Bayesian model averaging, a MAP estimator uses the single setting of weights (hypothesis) that maximizes the posterior density $w_{\text{MAP}} = \underset{w}{\arg\max}\, p(w|\mathcal{D}) = \underset{w}{\arg\max}(\log p(\mathcal{D}|w) + \log p(w))$, where the log prior can be viewed as a regularizer. For example, if we use a Gaussian prior on $w$, then $\log p(w)$ will penalize the $\ell_2$ norm of the parameters, driving parameters that do not improve the log likelihood $\log p(\mathcal{D}|w)$ to zero. MAP is the standard approach to training neural networks and our baseline for *classical training* throughout the paper. We perform MAP estimation with SGD unless otherwise specified.

**Covariate shift.** In this paper, we focus on the covariate shift setting. We assume the training dataset $\mathcal{D}_{\text{train}}$ consists of i.i.d. samples from the distribution $p_{\text{train}}(x, y) = p_{\text{train}}(x) \cdot p(y|x)$. However, the test data may come from a different distribution $p_{\text{test}}(x, y) = p_{\text{test}}(x) \cdot p(y|x)$. For concreteness, we assume the conditional distribution $p(y|x)$ remains unchanged, but the marginal distribution of the input features $p_{\text{test}}(x)$ differs from $p_{\text{train}}(x)$; we note that our results do not depend on this particular definition of covariate shift. Arjovsky [2021] provides a detailed discussion of covariate shift.

# 3   Related work

Methods to improve robustness to shift between train and test often explicitly make use of the test distribution in some fashion. For example, it is common to apply semi-supervised methods to the labelled training data augmented by the unlabelled test inputs [e.g., Daume III and Marcu, 2006, Athiwaratkun et al., 2018], or to learn a shared feature transformation for both train and test [e.g., Daumé III, 2009]. In a Bayesian setting, Storkey and Sugiyama [2007] and Storkey [2009] propose such approaches for linear regression and Gaussian processes under covariate shift, assuming the data comes from multiple sources. Moreover, Shimodaira [2000] propose to re-weight the train data points according to their density in the test data distribution. Sugiyama et al. [2006, 2007] adapt this importance-weighting approach to the cross-validation setting.

We focus on the setting of robustness to covariate shift without any access to the test distribution [e.g., Daume III and Marcu, 2006]. Bayesian methods are frequently applied in this setting, often motivated by the ability for a Bayesian model average to provide a principled representation of epistemic uncertainty: there are typically many consistent explanations for out of distribution points, leading to high uncertainty for these points. Indeed, approximate inference approaches for Bayesian neural networks are showing good and increasingly better performance under covariate shift [e.g., Gal and Ghahramani, 2016, Lakshminarayanan et al., 2017, Ovadia et al., 2019, Maddox et al., 2019, Wilson and Izmailov, 2020, Dusenberry et al., 2020, Benton et al., 2021].

Many works attempt to understand robustness to covariate shift. For example, for classical training Neyshabur et al. [2020] show that models relying on features that encode semantic structure in the data are more robust to covariate shift. Nagarajan et al. [2020] also provide insights into how classically trained max-margin classifiers can fail under covariate shift due to a reliance on spurious correlations between class labels and input features. BNN robustness to adversarial attacks is a related area of study, but generally involves much smaller perturbations to the test covariates than covariate shift. Carbone et al. [2020] prove that BNNs are robust to gradient-based adversarial attacks in the large data, overparameterized limit, while Wicker et al. [2021] present a framework for training BNNs with guaranteed robustness to adversarial examples.

In this work, we propose novel priors that improve robustness of BNNs under covariate shift. In Fortuin et al. [2021], the authors explore a wide range of priors and, in particular, show that the distribution of the weights of SGD-trained networks is heavy-tailed such as Laplace and Student-$t$, and does not appear Gaussian. Other heavy-tailed sparsity inducing priors have also been proposed

in the literature [Carvalho et al., 2009, Molchanov et al., 2017, Kessler et al., 2019, Cui et al., 2020, Izmailov et al., 2021, Fortuin, 2021]. Inspired by this work, we evaluate Laplace and Student-$t$ priors for BNNs, but find that they do not address the poor performance of BNNs under covariate shift.

Domingos [2000], Minka [2000], Masegosa [2019] and Morningstar et al. [2020] explore failure modes of Bayesian model averaging when the Bayesian model does not contain a reasonable solution in its hypothesis space, causing issues when the posterior contracts. This situation is orthogonal to the setting in our paper, where we know the Bayesian model does contain a reasonable solution in its hypothesis space, since the MAP estimate is robust to covariate shift. In our setting, robustness issues are caused by a *lack* of posterior contraction.

In general, understanding and addressing covariate shift is a large area of study. For a comprehensive overview, see Arjovsky [2021]. To our knowledge, no prior work has attempted to understand, further demonstrate, or remedy the poor robustness of Bayesian neural networks with high fidelity approximate inference recently discovered in Izmailov et al. [2021].

# 4  Bayesian neural networks are not robust to covariate shift

In this section, we evaluate Bayesian neural networks under different types of covariate shift. Specifically, we focus on two types of covariate shift: test data corruption and domain shift. In Appendix D, we additionally evaluate BNNs in the presence of spurious correlations in the data.

**Methods.**  We evaluate BNNs against two deterministic baselines: a MAP solution approximated with stochastic gradient descent (SGD) with momentum [Robbins and Monro, 1951, Polyak, 1964] and a deep ensemble of 10 independently trained MAP solutions [Lakshminarayanan et al., 2017]. For BNNs, we provide the results using a Gaussian prior and a more heavy-tailed Laplace prior following Fortuin et al. [2021]. Izmailov et al. [2021] conjectured that *cold posteriors* [Wenzel et al., 2020] can improve the robustness of BNNs under covariate shift; to test this hypothesis, we provide results for BNNs with a Gaussian prior and cold posteriors at temperature $10^{-2}$. For all BNN models, we run a single chain of HMC for 100 iterations discarding the first 10 iterations as burn-in, following Izmailov et al. [2021]. We provide additional experimental details in Appendix A.

**Datasets and data augmentation.**  We run all methods on the MNIST [LeCun et al., 2010] and CIFAR-10 [Krizhevsky et al., 2014] datasets. Following Izmailov et al. [2021] we do not use data augmentation with any of the methods, as it is not trivially compatible with the Bayesian neural network framework [e.g., Izmailov et al., 2021, Wenzel et al., 2020].

**Neural network architectures.**  On both the CIFAR-10 and MNIST datasets we use a small convolutional network (CNN) inspired by LeNet-5 [LeCun et al., 1998], with 2 convolutional layers followed by 3 fully-connected layers. On MNIST we additionally consider a fully-connected neural network (MLP) with 2 hidden layers of 256 neurons each. We note that high-fidelity posterior sampling with HMC is extremely computationally intensive. Even on the small architectures that we consider, the experiments take multiple hours on 8 NVIDIA Tesla V-100 GPUs or 8-core TPU-V3 devices [Jouppi et al., 2020]. See Izmailov et al. [2021] for details on the computational requirements of full-batch HMC for BNNs.

## 4.1  Test data corruption

We start by considering the scenario where the test data is corrupted by some type of noise. In this case, there is no semantic distribution shift: the test data is collected in the same way as the train data, but then corrupted by a generic transformation.

In Figure 2, we report the performance of the methods trained on MNIST and evaluated on the MNIST-corrupted (MNIST-C) test sets [Mu and Gilmer, 2019] under various corruptions[1]. We report the results for a fully-connected network and a convolutional network.

With the CNN architecture, deep ensembles consistently outperform BNNs. Moreover, even a single MAP solution significantly outperforms BNNs on many of the corruptions. The results are especially striking on *brightness* and *fog* corruptions, where the BNN with Laplace prior shows accuracy at the

---

[1]In addition to the corruptions from MNIST-C, we consider Gaussian noise with standard deviation 3, as this corruption was considered by Izmailov et al. [2021].

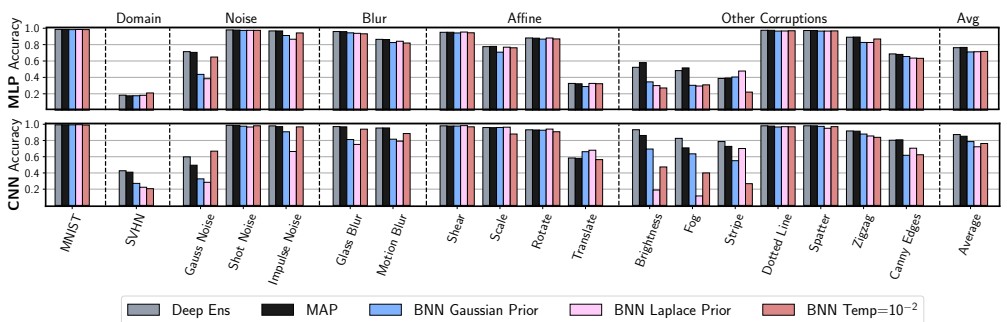

Figure 2: **Robustness on MNIST.** Accuracy for deep ensembles, MAP and Bayesian neural networks trained on MNIST under covariate shift. **Top**: Fully-connected network; **bottom**: Convolutional neural network. While on the original MNIST test set BNNs provide competitive performance, they underperform deep ensembles on most of the corruptions. With the CNN architecture, all BNN variants lose to MAP when evaluated on SVHN by almost 20%.

level of random guessing, while the deep ensemble retains accuracy above 90%. *Gaussian noise* and *impulse noise* corruptions also present significant challenges for BNNs. The BNNs show the most competitive performance in-distribution and on the corruptions representing affine transformations: *shear*, *scale*, *rotate* and *translate*. The results are generally analogous for the MLP: across the board the BNNs underperform deep ensembles, and often even a single MAP solution.

While the cold posteriors provide an improvement on some of the *noise* corruptions, they also hurt performance on *Brightness* and *Stripe*, and do not improve the performance significantly on average across all corruptions compared to a standard BNN with a Gaussian prior. We provide additional results on cold posterior performance in Appendix G.

Next, we consider the CIFAR-10-corrupted dataset (CIFAR-10-C) [Hendrycks and Dietterich, 2019]. CIFAR-10-C consists of 18 transformations that are available at different levels of intensity (1 − 5). We report the results using corruption intensity 4 (results for other intensities are in Appendix C) for each of the transformations in Figure 3. Similarly to MNIST-C, BNNs outperform deep ensembles and MAP on in-distribution data, but underperform each over multiple corruptions. On CIFAR-10-C, BNNs are especially vulnerable to different types of *noise* (*Gaussian noise*, *shot noise*, *impulse noise*, *speckle noise*). For each of the *noise* corruptions, BNNs underperform even the classical MAP solution. The cold posteriors improve the performance on the *noise* corruptions, but only provide a marginal improvement across the board.

We note that Izmailov et al. [2021] evaluated a ResNet-20 model on the same set of CIFAR-10-C corruptions. While they use a much larger architecture, the qualitative results for both architectures are similar: BNNs are the most vulnerable to *noise* and *blur* corruptions. We thus expect that our paper's analysis is not specific to smaller architectures and will equally apply to deeper models.

## 4.2 Domain shift

Next, we consider a different type of covariate shift where the test data and train data come from different, but semantically related distributions.

First, we apply our CNN and MLP MNIST models to the SVHN test set [Netzer et al., 2011]. The MNIST-to-SVHN domain shift task is a common benchmark for unsupervised domain adaptation: both datasets contain images of digits, although MNIST contains hand-written digits while SVHN represents house numbers. In order to apply our MNIST models to SVHN, we crop the SVHN images and convert them to grayscale. We report the results in Figure 2. While for MLPs all methods perform similarly, with the CNN architecture BNNs underperform deep ensembles and MAP by nearly 20%.

Next, we apply our CIFAR-10 CNN model to the STL-10 dataset [Coates et al., 2011]. Both datasets contain natural images with 9 shared classes between the two datasets[2]. We report the accuracy of

---

[2]CIFAR-10 has a class *frog* and STL-10 has *monkey*. The other nine classes coincide.

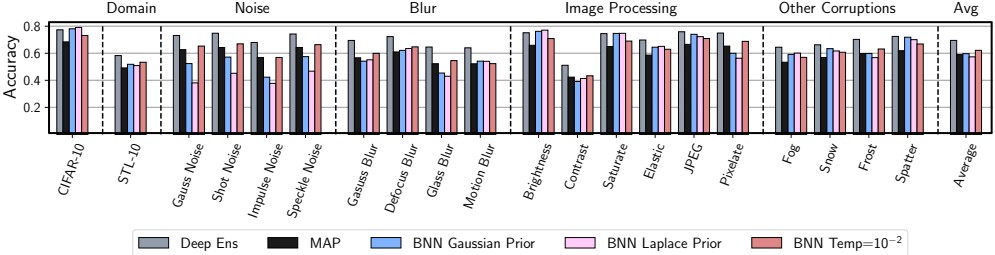

Figure 3: **Robustness on CIFAR-10.** Accuracy for deep ensembles, MAP and Bayesian neural networks using a CNN architecture trained on CIFAR-10 under covariate shift. For the corruptions from CIFAR-10-C, we report results for corruption intensity 4. While the BNNs with both Laplace and Gaussian priors outperform deep ensembles on the in-distribution accuracy, they underperform even a single MAP solution on most corruptions.

the CIFAR-10 models on these 9 shared classes in STL-10 in Figure 3. While BNNs outperform the MAP solution, they still significantly underperform deep ensembles.

The results presented in this section highlight the generality and practical importance of the lack of robustness in BNNs: despite showing strong performance in-distribution, BNNs underperform even a single MAP solution (classical training) over an extensive variety of covariate shifts.

## 5 Understanding Bayesian neural networks under covariate shift

Now that we have established that Bayesian neural networks are highly susceptible to many types of covariate shift, we seek to understand why this is the case. In this section, we identify the linear dependencies in the input features as one of the key issues undermining the robustness of BNNs. We emphasize that *linear* dependencies in particular are *not* simply chosen for the simplicity of analysis, and their key role follows from the structure of the fully-connected and convolutional layers.

### 5.1 Motivating example: dead pixels and fully-connected layers

To provide an intuition for the results presented in this section, we start with a simple but practically relevant motivating example. Suppose we use a fully-connected Bayesian neural network on a dataset $D = \{(x_i, y_i)\}_{i=1}^n$, with $m$ input features, where $x_i \in \mathbb{R}^m$. We use the upper indices to denote the features $x^1, \ldots, x^m$ and the lower indices to denote the datapoints $x_i$. Then, for each neuron $j$ in the first hidden layer of the network, the activation can be written as $z_j = \phi(\sum_{i=1}^n x^i w_{ij}^1 + b_j^1)$, where $w_{ij}^1$ is the weight of the first layer of the network corresponding to the input feature $i$ and hidden neuron $j$, and $b_j^1$ is the corresponding bias. We show the following Lemma:

**Lemma 1** *Using the notation introduced above, suppose that the input feature $x_k^i$ is equal to zero for all the examples $x_k$ in the training dataset $D$. Suppose the prior distribution over the parameters $p(W)$ factorizes as $p(W) = p(w_{ij}^1) \cdot p(W \setminus w_{ij}^1)$ for some neuron $j$ in the first layer, where $W \setminus w_{ij}^1$ represents all the parameters $W$ of the network except $w_{ij}^1$. Then, the posterior distribution $p(W|D)$ will also factorize and the marginal posterior over the parameter $w_{ij}^1$ will coincide with the prior:*

$$p(W|D) = p(W \setminus w_{ij}^1|D) \cdot p(w_{ij}^1). \tag{3}$$

*Consequently, the MAP solution will set the weight $w_{ij}^1$ to the value with maximum prior density.*

Intuitively, Lemma 1 says that if the prior for some parameter in the network is independent of the other parameters, and the value of the parameter does not affect the predictions of the model on any of the training data, then the posterior of this parameter will coincide with its prior. In particular, if one of the input features is always zero, then the corresponding weights will always be multiplied by zero, and will not affect the predictions of the network. To prove Lemma 1, we simply note that the posterior is proportional to the product of prior and likelihood, and both terms factorize with respect to $w_{ij}^1$. We present a formal proof in Appendix H.

So, for any sample from the posterior, the weight $w_{ij}^1$ will be a random draw from the prior distribution. Now suppose at test time we evaluate the model on data where the feature $x^i$ is no longer zero. Then for these new inputs, the model will be effectively multiplying the input feature $x^i$ by a random weight $w_{ij}^1$, leading to instability in predictions. In Appendix H, we formally state and prove the following proposition:

**Proposition 1 (informal)** *Suppose the assumptions of Lemma 1 hold. Assume also that the prior distribution $p(w_{ij}^1)$ has maximum density at 0 and that the network uses ReLU activations. Then for any test input $\bar{x}$, the expected prediction under Bayesian model averaging (Equation 2) will depend on the value of the feature $\bar{x}^i$, while the MAP solution will ignore this feature.*

For example, in the MNIST dataset there is a large number of *dead pixels*: pixels near the boundaries of the image that have intensity zero for all the inputs. In practice, we often use independent zero-mean priors (e.g. Gaussian) for each parameter of the network. So, according to Lemma 1, the posterior over all the weights in the first layer of the network corresponding to the dead pixels will coincide with the prior. If at test time the data is corrupted by e.g., Gaussian noise, the dead pixels will receive non-zero intensities, leading to a significant degradation in the performance of the Bayesian model average compared to the MAP solution.

While the situation where input features are constant and equal to zero may be rare and easily addressed, the results presented in this section can be generalized to *any* linear dependence in the data. We will now present our results in the most general form.

## 5.2  General linear dependencies and fully-connected layers

We now present our general results for fully-connected Bayesian neural networks when the features are linearly dependent. Intuitively, if there exists a direction in the input space such that all of the training data points have a constant projection on this direction (i.e. the data lies in a hyper-plane), then posterior coincides with the prior in this direction. Hence, the BMA predictions are highly susceptible to perturbations that move the test inputs in a direction orthogonal to the hyper-plane. The MAP solution on the other hand is completely robust to such perturbations. In Appendix H we prove the following proposition.

**Proposition 2** *Suppose that the prior over the weights $w_{ij}^1$ and biases $b_j^1$ in the first layer is an i.i.d. Gaussian distribution $\mathcal{N}(0, \alpha^2)$, independent of the other parameters in the model. Suppose all the inputs $x_1 \ldots x_n$ in the training dataset $D$ lie in an affine subspace of the input space: $\sum_{j=1}^m x_i^j c_j = c_0$ for all $i = 1, \ldots, n$ and some constants $c$ such that $\sum_{i=0}^m c_i^2 = 1$. Then,*

*(1) For any neuron $j$ in the first hidden layer, the posterior distribution of random variable $w_j^c = \sum_{i=1}^m c_i w_{ij}^1 - c_0 b_j^1$ (the projection of parameter vector $(w_{1j}^1, \ldots, w_{mj}^1, b_j^1)$ on direction $(c_1, \ldots, c_m, -c_0)$) will coincide with the prior $\mathcal{N}(0, \alpha^2)$.*

*(2) The MAP solution will set $w_j^c$ to zero.*

*(3) (Informal) Assuming the network uses ReLU activations, at test time, the BMA prediction will be susceptible to the inputs $\bar{x}$ that lie outside of the subspace, i.e. the predictive mean will depend on $\sum_{j=1}^m \bar{x}^j c_j - c_0$. The MAP prediction will not depend on this difference.*

**Empirical support.**    To test Proposition 2, we examine the performance of a fully-connected BNN on MNIST. The MNIST training dataset is not full rank, meaning that it has linearly dependent features. For a fully-connected BNN, Proposition 2 predicts that the posterior distribution of the first layer weights projected onto directions corresponding to these linearly dependent features will coincide with the prior. In Figure 4(a) we test this hypothesis by projecting first layer weights onto the principal components of the data. As expected, the distribution of the projections on low-variance PCA components (directions that are constant or nearly constant in the data) almost exactly coincides with the prior. The MAP solution, on the other hand, sets the weights along these PCA components close to zero, confirming conclusion (2) of the proposition. Finally, in Figure 4(b) we visualize the performance of the BMA and MAP solution as we apply noise along high-variance and low-variance directions in the data. As predicted by conclusion (3) of Proposition 2, the MAP solution is very robust to noise along the low-variance directions, while BMA is not.

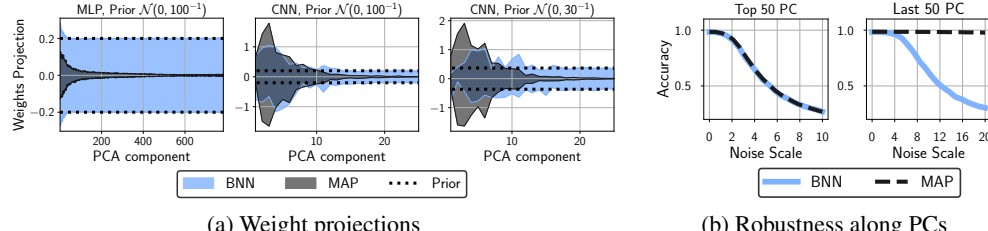

(a) Weight projections  (b) Robustness along PCs

Figure 4: **Bayesian inference samples weights along low-variance principal components from the prior, while MAP sets these weights to zero.** **(a)**: The distribution (mean $\pm$ 2 std) of projections of the weights of the first layer on the directions corresponding to the PCA components of the data for BNN samples and MAP solution using MLP and CNN architectures with different prior scales. In each case, MAP sets the weights along low-variance components to zero, while BNN samples them from the prior. **(b)**: Accuracy of BNN and MAP solutions on the MNIST test set with Gaussian noise applied along the 50 highest and 50 lowest variance PCA components of the train data (left and right respectively). MAP is very robust to noise along low-variance PCA directions, while BMA is not; the two methods are similarly robust along the highest-variance PCA components.

## 5.3 Linear dependencies and convolutional layers

Finally, we can extend Proposition 2 to convolutional layers. Unlike fully-connected layers, convolutional layers include weight sharing such that no individual weight corresponds to a specific pixel in the input images. For example, dead pixels will not necessarily present an issue for convolutional layers, unlike what is described in subsection 5.1. However, convolutional layers are still susceptible to a special type of linear dependence. Intuitively, we can think of the outputs of the first convolutional layer on all the input images as the outputs of a fully-connected layer applied to all the $K \times K$ patches of the input image. Therefore the reasoning in Proposition 2 applies to the convolutional layers, with the difference that the linear dependencies in the $K \times K$ patches cause the instability rather than dependencies in the full feature space. In Appendix H we prove the following proposition.

**Proposition 3** *Suppose that the prior over the parameters of the convolutional filters and biases in the first layer is an i.i.d. Gaussian distribution $\mathcal{N}(0, \alpha^2)$, independent of the other parameters in the model. Suppose that the convolutional filters in the first layer are of size $K \times K \times C$, where $C$ is the number of input channels. Then, consider the set $\hat{D}$ of size $N$ of all the patches of size $K \times K \times C$ extracted from the training images in $D$ after applying the same padding as in the first convolutional layer. Suppose all the patches $z_1 \ldots z_N$ in the dataset $\hat{D}$ lie in an affine subspace of the space $\mathbb{R}^{K \times K \times C}$: $\sum_{c=1}^{C} \sum_{a=1}^{K} \sum_{b=1}^{K} z_i^{a,b} \gamma_{c,a,b} = \gamma_0$ for all $i = 1, \ldots, N$ and some constants $c_i$ such that $\sum_{c=1}^{C} \sum_{a=1}^{K} \sum_{b=1}^{K} \gamma_{c,a,b}^2 + \gamma_0^2 = 1$. Then, we can prove results analogous to (1)-(3) in Proposition 2 (see the Appendix H for the details).*

**Empirical support.** In Figure 4(a), we visualize the projections of the weights in the first layer of the CNN architecture on the PCA components of the $K \times K$ patches extracted from MNIST. Analogously to fully-connected networks, the projections of the MAP weights are close to zero for low-variance components, while the projections of the BNN samples follow the prior.

## 5.4 What corruptions will hurt performance?

Based on Propositions 2, 3, we expect that the corruptions that are the most likely to break linear dependence structure in the data will hurt the performance the most. In Appendix I we argue that *noise* corruptions are likely to break linear dependence, while the *affine* corruptions are more likely to preserve it, agreeing with our observations in section 4.

## 5.5 Why do some approximate Bayesian inference methods work well under covariate shift?

Unlike BNNs with HMC inference, some approximate inference methods such as SWAG [Maddox et al., 2019], MC dropout [Gal and Ghahramani, 2016], deep ensembles [Lakshminarayanan et al., 2017] and mean field VI [Blundell et al., 2015] provide strong performance under covariate shift

[Ovadia et al., 2019, Izmailov et al., 2021]. For deep ensembles, we can easily understand why: a deep ensemble represents an average of approximate MAP solutions, and we have seen in conclusion (3) of Proposition 2 that MAP is robust to covariate shift in the scenario introduced in subsection 5.2. Similarly, other methods are closely connected to MAP via characterizing the posterior using MAP optimization iterates (SWAG), or modifying the training procedure for the MAP solution (MC Dropout). We provide further details, including theoretical and empirical analysis of variational inference under covariate shift, in Appendix J.

# 6 Towards more robust Bayesian model averaging under covariate shift

In this section, we propose a simple new prior inspired by our theoretical analysis. In section 5 we showed that linear dependencies in the input features cause the posterior to coincide with the prior along the corresponding directions in the parameter space. In order to address this issue, we explicitly design the prior for the first layer of the network so that the variance is low along these directions.

## 6.1 Data empirical covariance prior

Let us consider the empirical covariance matrix of the inputs $x_i$. Assuming the input features are all preprocessed to be zero-mean $\sum_{i=1}^n x_i = 0$, we have $\Sigma = \frac{1}{n-1} \sum_{i=1}^n x_i x_i^T$. For fully-connected networks, we propose to use the *EmpCov* prior $p(w^1) = \mathcal{N}(0, \alpha\Sigma + \epsilon I)$ on the weights $w^1$ of the first layer of the network, where $\epsilon$ is a small positive constant ensuring that the covariance matrix is positive definite. The parameter $\alpha > 0$ determines the scale of the prior.

Suppose there is a linear dependence in the input features of the data: $x_i^T p = c$ for some direction $p$ and constant $c$. Then $p$ will be an eigenvector of the empirical covariance matrix with the corresponding eigenvalue equal to $0$ : $\Sigma p = \frac{1}{n-1} \sum_{i=1}^n x_i x_i^T p = \frac{c}{n-1} \sum_{i=1}^n x_i = 0$. Hence the prior over $w^1$ will have a variance of $\epsilon$ along the direction $p$.

More generally, the *EmpCov* prior is aligned with the principal components of the data, which are the eigenvectors of the matrix $\Sigma$. The prior variance along each principal component $p_i$ is equal to $\alpha\sigma_i^2 + \epsilon$ where $\sigma_i^2$ is its corresponding explained variance. In Appendix K, we discuss a more general family of priors aligned with the principal components of the data.

**Generalization to convolutions.** We can generalize the *EmpCov* prior to convolutions by replacing the empirical covariance of the data with the empirical covariance of the patches that interact with the convolutional filter, denoted by $\hat{D}$ in Proposition 3.

**Is EmpCov a valid prior?** *EmpCov* constructs a valid prior by evaluating the empirical covariance matrix of the inputs. This prior does not depend on the train data labels $y_i$, unlike the approach known as *Empirical Bayes* [see e.g. Bishop, 2006, section 3.5], which is commonly used to specify hyperparameters in Gaussian process and neural network priors [Rasmussen and Williams, 2006, MacKay, 1995].

## 6.2 Experiments

In Figure 5 we report the performance of the BNNs using the *EmpCov* prior. In each case, we apply the *EmpCov* prior to the first layer, and a Gaussian prior to all other layers. For more details, please see Appendix A. On both MLP on MNIST and CNN on CIFAR-10, the *EmpCov* prior significantly improves performance across the board. In both cases, the BNN with *EmpCov* prior shows competitive performance with deep ensembles, especially in the domain shift experiments. *EmpCov* is also particularly useful on the *noise* corruptions.

In Appendix L, we provide a detailed analysis of the performance of the CNN architecture on MNIST. Surprisingly, we found that using the *EmpCov* prior by itself does not provide a large improvement in this case. In Appendix L, we identify an issue specific to this particular setting, and propose another targeted prior that substantially improves performance.

# 7 Discussion

We consider the generality of our results, additional perspectives, and future directions.

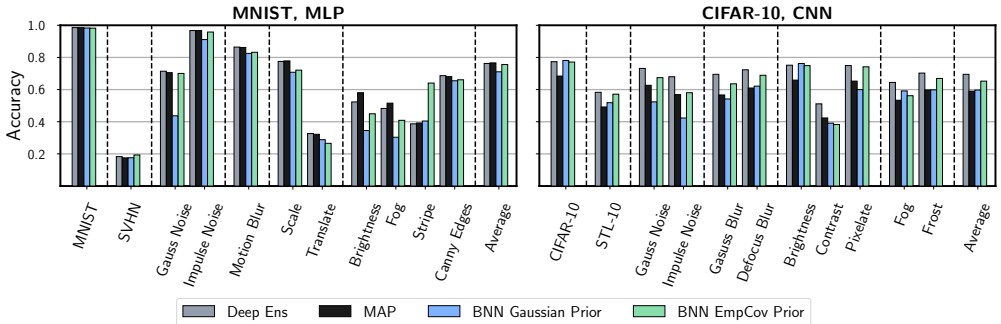

Figure 5: **EmpCov prior improves robustness.** Test accuracy under covariate shift for deep ensembles, MAP optimization with SGD, and BNN with Gaussian and *EmpCov* priors. **Left**: MLP architecture trained on MNIST. **Right**: CNN architecture trained on CIFAR-10. The *EmpCov* prior provides consistent improvement over the standard Gaussian prior. The improvement is particularly noticeable on the *noise* corruptions and domain shift experiments (SVHN, STL-10).

**Is the poor generalization of BNNs under covariate shift surprising?** The results presented in this paper are of crucial practical importance, relevant to the applicability of Bayesian neural networks in virtually every real-world setting since the train distribution is often not exactly the same as the test distribution. Ultimately, whether or not a result is *surprising* is subjective, but there are many reasons to find the dramatic performance degradation of Bayesian neural networks under shift surprising, which we outline in Appendix B.

**Why focus on linear dependencies?** This focus is dictated by the structure of both fully-connected and convolutional neural networks, which apply activation functions to linear combinations of features. Our results can be directly extended to other models, where different types of dependencies between input features would lead to the same lack of robustness to covariate shift. For example, in Appendix M we derive analogous results to subsection 5.2 for multiplicative neural network architectures [Trask et al., 2018], where a different form of dependence between features causes poor robustness.

**Intermediate layers.** While our analysis in section 5 is focused on the linear dependencies in the input features, similar conclusions can be made about intermediate layers of the network. For example, weights connected to *dead neurons*, which output zero, do not affect predictions of the model. We provide more details in Appendix N.

**Linear Bayesian models.** In models that are linear in parameters, such as Bayesian linear regression and Gaussian process regression, we are typically saved from the perils of BMA under covariate shift discussed in Section 5, because the MAP and the predictive mean under BMA coincide. We provide further details in Appendix O.

**BNNs in low-data regime.** While we focus on covariate shift, our results also suggest that BNNs may generalize poorly when the training dataset is extremely small. Indeed, in low-data regime we may observe linear combinations of the features that are constant in the training data, but not on test. In Appendix E we show empirically that BNNs can underperform MAP in low-data regime.

**An optimization perspective on the covariate shift problem.** In Section 5.1, we argued that SGD is more robust to covariate shift than HMC because the regularizer pushes the weights that correspond to dead pixels towards zero. This effect is mainly obtained through explicit regularization, without which these weights will remain at their initial values. We study the effect of initialization and explicit regularization on the performance of SGD under covariate shift in Appendix P.1. We also show in Appendix P.2 that other stochastic optimizers such as Adam [Kingma and Ba, 2014] and Adadelta [Zeiler, 2012] behave similarly to SGD under covariate shift. Finally, we discuss the effect of test data corruptions on the loss landscape in Appendix P.3 and argue that the relative sharpness of low density posterior samples makes these solutions, which are included in a BMA but not MAP, more vulnerable to covariate shift.

**Limitations.** Due to the intense computational requirements of HMC, our experiments are limited to smaller models and datasets. Our analysis is focused on issues arising in models that are non-linear

in their parameters. Moreover, while our proposed priors help improve robustness, they do not entirely resolve the issue. For example, in the *Contrast* dataset of Figure 5 (right panel), the BMA is still underperforming MAP.

**Conclusion.** Our work has demonstrated, both empirically and theoretically, how linear dependencies in the training data cause Bayesian neural networks to generalize poorly under covariate shift — explaining the important and unexpected findings in Izmailov et al. [2021]. The scope of this research is exceptionally broad, relevant to the safe deployment of Bayesian methods in virtually any real-world setting. While the two priors we introduce achieve some improvement in BNN performance under covariate shift, we are only beginning to explore possible remedies. Our work is intended as a step towards understanding the true properties of Bayesian neural networks, and improving the robustness of Bayesian model averaging under covariate shift.

## Acknowledgements

We thank Martin Arjovsky, Behnam Neyshabur, Vaishnavh Nagarajan, Marc Finzi, Polina Kirichenko, Greg Benton and Nate Gruver for helpful discussions. This research is supported with Cloud TPUs from Google's TPU Research Cloud (TRC), and by an Amazon Research Award, NSF I-DISRE 193471, NIH R01DA048764-01A1, NSF IIS-1910266, and NSF 1922658 NRT-HDR: FUTURE Foundations, Translation, and Responsibility for Data Science.

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
