## Appendix Outline

This appendix is organized as follows.

- In Appendix A, we provide details on hyper-parameters, datasets and architectures used in our experiments.
- In Appendix B, we discuss whether the poor generalization of BNNs under covariate shift is surprising.
- In Appendix C, we examine BNN performance under covariate shift for a variety of different standard priors with different hyper-parameter settings.
- In Appendix D, we study the effect of spurious correlations on BMA performance using the shift-MNIST dataset.
- In Appendix E, we show that the same issues that hurt BNN generalization under covariate shift can cause poor performance in low-data regime.
- In Appendix F, we explore the convergence of the BNN performance as a function of the number of HMC samples we produce.
- In Appendix G, we study how temperature scaling impacts BMA performance under covariate shift.
- In Appendix H, we provide proofs of our propositions from section 5.
- In Appendix I, we visualize how different corruptions introduce noise along different principal components of the data, and relate this to BMA performance on these corruptions.
- In Appendix J, we explain why approximate inference methods SWAG and MC Dropout do not suffer the same performance degradation under covariate shift as HMC.
- In Appendix K, we analyze a more general family of priors that includes the *EmpCov* prior from section 6.
- In Appendix L, we introduce the sum filter prior for improving BNN robustness to non-zero-mean noise.
- In Appendix M, we provide an example of a model architecture where the BMA will be impacted by nonlinear dependencies in the training data.
- In Appendix N, we examine how BNNs can be impacted by linear dependencies beyond the first layer using the example of dead neurons.
- In Appendix O we prove that linear dependencies do not hurt BMAs of linear models under covariate shift.
- In Appendix P, we examine covariate shift from an optimization perspective.
- Lastly, in Appendix Q, we provide details on licensing.

## A   Hyper-parameters and details of experiments

### A.1   Prior definitions

Here we define the prior families used in the main text and the appendix, and the corresponding hyper-parameters.

**Gaussian priors.**     We consider iid Gaussian priors of the form $\mathcal{N}(0, \alpha^2 I)$, where $\alpha^2$ is the prior variance. Gaussian priors are the default choice in Bayesian neural networks [e.g. 21, 28, 69].

**Laplace priors.**     We consider priors of the form $\text{Laplace}(\alpha) : \frac{1}{2\alpha} \exp(-\|x\|_1/\alpha)$, where $\|\cdot\|_1$ is the $\ell_1$-norm.

**Student-t priors.**     In Appendix C, we consider iid Student-t priors of the form $\text{Student-}t(\nu, \alpha^2) : \frac{\Gamma(\frac{\nu+1}{2})}{\Gamma(\frac{\nu}{2})\sqrt{\nu\pi}}(1 + \frac{w^2}{\nu\alpha^2})^{-\frac{\nu+1}{2}}$, where $\nu$ represents the degrees of freedom and $\alpha^2$ is the prior variance.

**Exp-norm priors.**     In Appendix C, we consider the prior family of the form $\text{ExpNorm}(p, \alpha^2) : \exp(-\|w\|^p/2\alpha^2)$. Notice that for $p = 2$, we get the Gaussian prior family. By varying $p$ we can construct more heavy-tailed ($p < 2$) or less heavy-tailed ($p > 2$) priors.

## A.2 Hyper-parameters and details

**HMC hyper-parameters.** The hyper-parameters for HMC are the step size, trajectory length and any hyper-parameters of the prior. Following Izmailov et al. [28], we set the trajectory length $\tau = \frac{\pi \sigma_{prior}}{2}$ where $\sigma_{prior}$ is the standard deviation of the prior. We choose the step size to ensure that the accept rates are high; for most of our MLP runs we do $10^4$ leapfrog steps per sample, while for CNN we do $5 \cdot 10^3$ leapfrog steps per sample. For each experiment, we run a single HMC chain for 100 iterations discarding the first 10 iterations as burn-in; in Appendix F we show that 100 samples are typically sufficient for convergence of the predictive performance.

**Data Splits.** For all CIFAR-10 and MNIST experiments, we use the standard data splits: 50000 training samples for CIFAR-10, 60000 training samples for MNIST, and 10000 test samples for both. For all data corruption experiments, we evaluate on the corrupted 10000 test samples. For domain shift experiments, we evaluate on 26032 SVHN test samples for MNIST to SVHN and 7200 STL-10 test samples for CIFAR-10 to STL-10. In all cases, we normalize the inputs using train data statistics, and do not use any data augmentation.

**Neural network architectures.** Due to computational constraints, we use smaller neural network architectures for our experiments. All architectures use ReLU activations. For MLP experiments, we use a fully-connected network with 2 hidden layers of 256 neurons each. For CNN experiments, we use a network with 2 convolutional layers followed by 3 fully-connected layers. Both convolutional layers have $5 \times 5$ filters, a stride of 1, and use $2 \times 2$ average pooling with stride 2. The first layer has 6 filters and uses padding, while the second layer has 16 filters and does not use padding. The fully connected layers have 400, 120, and 84 hidden units.

**MAP and deep ensemble hyper-parameters.** We use the SGD optimizer with momentum 0.9, cosine learning rate schedule and weight decay 100 to approximate the MAP solution. In Appendix P we study the effect of using other optimizers and weight decay values. On MNIST, we run SGD for 100 epochs, and on CIFAR we run for 300 epochs. For the deep ensemble baselines, we train 10 MAP models independently and ensemble their predictions.

**Prior hyper-parameters.** To select prior hyper-parameters we perform a grid search, and report results for the optimal hyperparameters in order to compare the best versions of different models and priors. We report the prior hyper-parameters used in our main evaluation in Table 1. In Appendix C we provide detailed results for various priors with different hyper-parameter choices.

Table 1: Prior hyper-parameters

| Hyper-parameter | MNIST MLP | MNIST CNN | CIFAR-10 CNN |
|---|---|---|---|
| BNN, Gaussian prior; $\alpha^2$ | $\frac{1}{100}$ | $\frac{1}{100}$ | $\frac{1}{100}$ |
| BNN, Laplace prior; $\alpha$ | $\sqrt{\frac{1}{6}}$ | $\sqrt{\frac{1}{6}}$ | $\sqrt{\frac{1}{200}}$ |

**Tempering hyper-parameters.** For the tempering experiments, we use a Gaussian prior with variance $\alpha^2 = \frac{1}{100}$ on MNIST and $\alpha^2 = \frac{1}{3}$ on CIFAR-10. We set the posterior temperature to $10^{-2}$. We provide additional results for other prior variance and temperature combinations in Appendix G.

**Compute.** We ran all the MNIST experiments on TPU-V3-8 devices, and all CIFAR experiments on 8 NVIDIA Tesla-V100 devices. A single HMC chain with 100 iterations on these devices takes roughly 1.5 hours for MNIST MLP, 2 hours for CIFAR CNN and 3 hours for MNIST CNN. As a rough upper-bound, we ran on the order of 100 different HMC chains, each taking 2 hours on average, resulting in 200 hours on our devices, or roughly 1600 GPU-hours (where we equate 1 hour on TPU-V3-8 to 8 GPU-hours).

# B  Should we find the lack of BMA robustness surprising?

Bayesian neural networks are sometimes presented as a way of improving *just* the uncertainties, often at the cost of degradation in accuracy. Consequently, one might assume that the poor performance of BNNs under covariate shift is not surprising, and we should use BNN uncertainty estimates

solely to *detect* OOD, without attempting to make predictions, even for images that are still clearly recognizable.

In recent years, however, Bayesian deep learning methods [e.g., 43, 17, 15], as well as high-fidelity approximate inference with HMC [28], achieve improved uncertainty *and* accuracy compared to standard MAP training with SGD. In this light, we believe there are many reasons to find the significant performance degradation under shift surprising:

- The BNNs are often providing significantly better accuracy on in-distribution points. For example, HMC BNNs achieve a 5% improvement over MAP on CIFAR-10, but 25% worse accuracy on the pixelate corruption, when the images are still clearly recognizable (see Figure 1). To go from clearly better to profoundly worse would not typically be expected of any method on these shifts.

- In fact, recent work [e.g. 46] shows that there is typically a strong correlation between in-distribution and OOD generalization accuracy on related tasks, which is the opposite of what we observe in this work.

- Many approximate Bayesian inference procedures do improve accuracy over MAP on shift problems [56, 69, 17], and newer inference procedures appear to be further improving on these results. For example, MultiSWAG [69] is significantly more accurate than MAP under shift. The fact that these methods are more Bayesian than MAP, and improve upon MAP in these settings, makes it particularly surprising that a high-fidelity BMA would be so much worse than MAP. This is a nuanced point — how is it that methods getting closer in some ways to the Bayesian ideal are improving on shift, when a still higher-fidelity representation of the Bayesian ideal is poor on shift? — we discuss this point in Appendix J.

- Recent results highlight that there need not be a tension between OOD detection and OOD generalization accuracy: indeed deep ensembles provide much better performance than MAP on both [56].

- Bayesian methods are closely associated with trying to provide a good representation of uncertainty, and a good representation of uncertainty should not say "I have little idea" when a point is only slightly out of distribution, but still clearly recognizable, e.g., through noise corruption or mild domain shift.

- In Figure 6 we report the log-likelihood and ECE metrics which evaluate the quality of uncertainty estimates for deep ensembles, MAP and BNNs. The log-likelihood and ECE of standard BNNs are better than the corresponding values for the MAP solution on average, but they are much worse than the corresponding numbers for deep ensembles for high degrees of corruption. Furthermore, for some corruptions (*impulse noise*, *pixelate*) BNNs lose to MAP on both log-likelihood and ECE at corruption intensity 5. Also for larger ResNet-20 architecture on CIFAR-10-C, Izmailov et al. [28] reported that the log-likelihoods of BNNs are on average slightly worse than for MAP solution at corruption intensity 5.

## C   Additional results on BNN robustness

### C.1   Error-bars and additional metrics

We report the accuracy, log-likelihood and expected calibration error (ECE) for deep ensembles, MAP solutions and BMA variations in Figure 6. We report the results for different corruption intensities (1, 3, 5) and provide error-bars computed over 3 independent runs. Across the board, *EmpCov* priors provide the best performance among BNN variations on all three metrics.

### C.2   Detailed results for different priors

In this section, we evaluate BNNs with several prior families and provide results for different choices of hyper-parameters. The priors are defined in subsection A.1.

We report the results using CNNs and MLPs on MNIST in Figure 7. None of the considered priors completely close the gap to MAP under all corruptions. Gaussian priors show the worst results, losing to MAP on all MNIST-C corruptions and Gaussian noise, at all prior standard deviations. Laplace priors show similar results to Gaussian priors under Gaussian noise, but beat MAP on the *stripe*

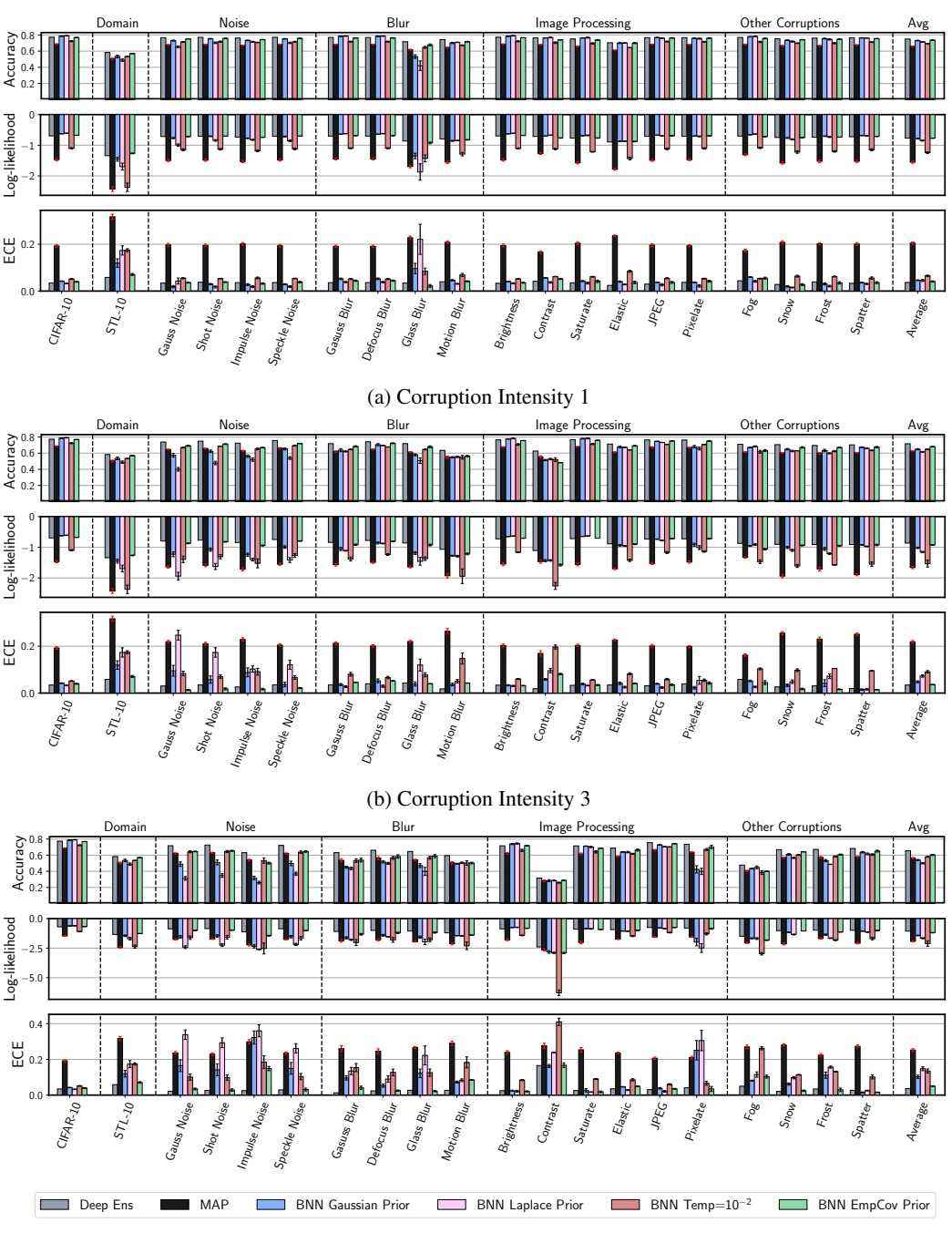

(a) Corruption Intensity 1

(b) Corruption Intensity 3

(c) Corruption Intensity 5

Figure 6: **Detailed results on CIFAR-10.** Accuracy, log-likelihood and log-likelihood for deep ensembles, MAP solution, and BNN variants under covariate shift on CIFAR-10. We report the performance at corruption intensity levels 1, 3 and 5 (corruption intensity does not affect the CIFAR-10 and STL-10 columns in the plots). For all methods except deep ensembles we report the mean and standard deviation (via error-bars) over 3 random seeds. *EmpCov* priors provide the only BNN variation that consistently performs on par with deep ensembles in terms of log-likelihood and ECE. Tempered posteriors improve the accuracy on some of the corruptions, but significantly hurt in-domain performance.

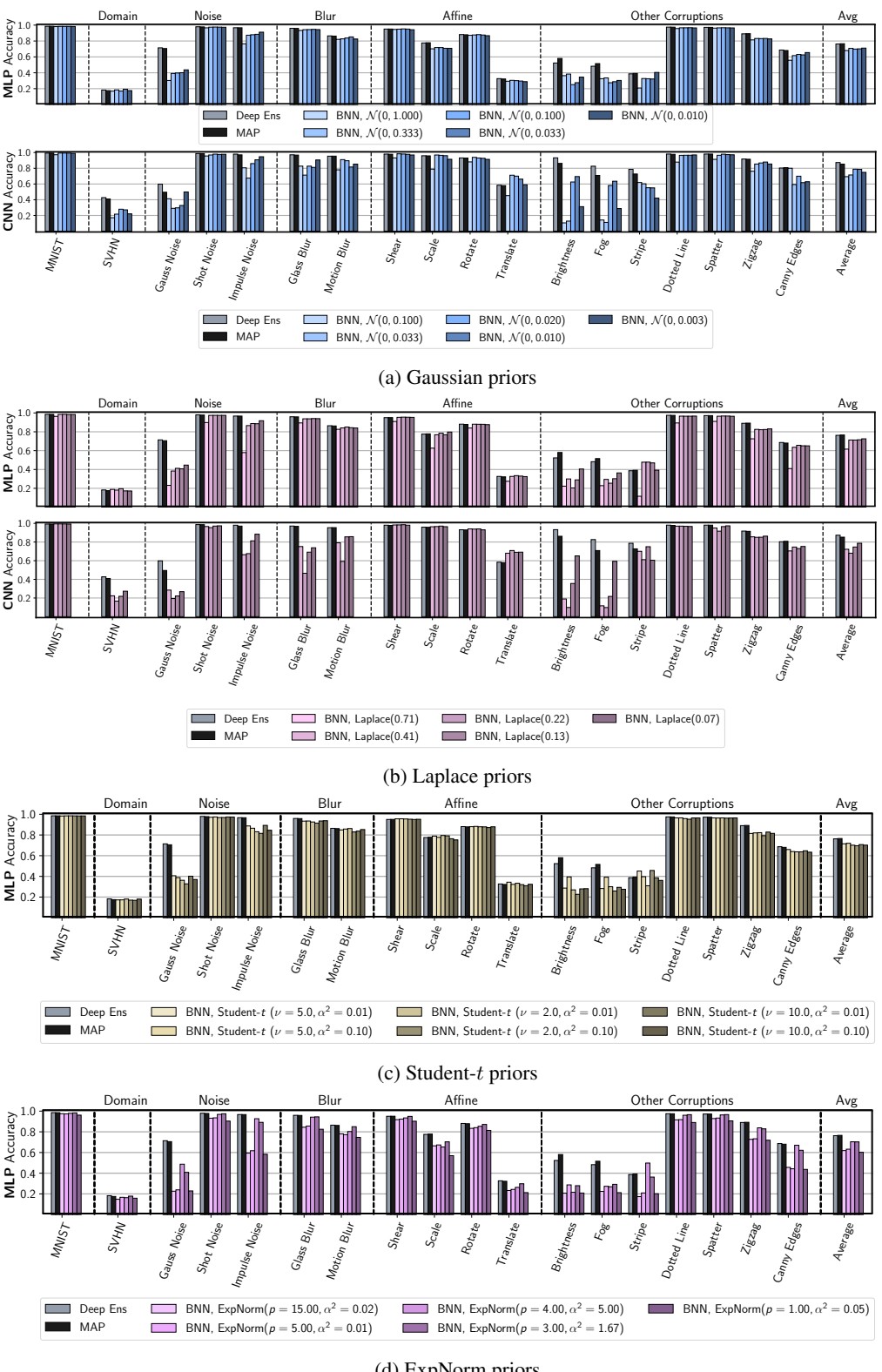

Figure 7: **Priors on MNIST.** We report the performance of different prior families under covariate shift on MNIST. For Gaussian and Laplace prior families we report the results using both the MLP and CNN architectures; for Student-$t$ and ExpNorm we only report the results for MLP. None of the priors can match the MAP performance across the board, with particularly poor results under *Gaussian noise*, *Brightness* and *Fog* corruptions.

corruption in MNIST-C. Student-$t$ priors show better results, matching or outperforming MAP on all affine transformations, but still underpeform significantly under *Gaussian noise*, *Brightness* and *Fog* corruptions. Finally, exp-norm priors can match MAP on Shot noise and also outperform MAP on *stripe*, but lose on other corruptions. The in-domain performance with exp-norm priors is also lower compared to the other priors considered.

To sum up, none of the priors considered is able to resolve the poor robustness of BNNs under covariate shift. In particular, all priors provide poor performance under *Gaussian noise*, *Brightness* and *Fog* corruptions.

## D    Bayesian neural networks and spurious correlations

For corrupted data, models experience worse performance due to additional noisy features being introduced. However, it's also possible that the reverse can occur, and a seemingly highly predictive feature in the training data will not be present in the test data. This distinct category of covariate shift is often called spurious correlation. To test performance with spurious correlations, we use the Shift-MNIST dataset [29], where we introduce spurious features via modifying the training data so that a set of ten pixels in the image perfectly correlates with class labels.

Table 2: **Spurious correlations.** Accuracy and log-likelihood of MAP, deep ensembles and BNNs with Gaussian and *EmpCov* priors on the Shift-MNIST dataset.

| Model | MLP Accuracy | MLP LL | CNN Accuracy | CNN LL |
|---|---|---|---|---|
| MAP | 88.70% | -0.527 | 48.63% | -2.206 |
| Deep Ensemble | 88.73% | -0.527 | 72.99% | -1.041 |
| BNN Gaussian Prior | 90.83% | -0.598 | 64.27% | -1.326 |
| BNN EmpCov Prior | 86.95% | -1.146 | 64.41% | -1.450 |

Table 2 shows the results for deep ensembles, MAP and BNNs with Gaussian and *EmpCov* priors on the Shift-MNIST dataset. We see worse accuracy for CNN architectures, demonstrating how more complex architectures can more easily over-fit the spurious correlations. BNNs with Gaussian prior perform better than MAP on both MLP and CNN, but significantly worse than deep ensembles for CNNs. Notice that the *EmpCov* prior does not improve performance here for either architecture, highlighting the difference between spurious correlations and other forms of covariate shift. In particular, the largest principal components of the Shift-MNIST training dataset place large magnitude weights on the spurious features, and so using the *EmpCov* prior results in samples with larger weights for the activated (spurious) pixels. When those same pixels are not activated in the test set, such samples will have a larger shift in their predictions.

An in-depth analysis of BNNs in the presence of spurious correlations remains an exciting direction for further research.

## E    Bayesian neural networks in low-data regime

The intuition presented in Propositions 2, 3 suggests that Bayesian neural networks may also under-perform in low-data regime. Indeed, if the model only observes a small number of datapoints, some of the directions in the parameter space will not be sufficiently constrained by the data. Empirically, in Table 3 found that the performance of BNNs is indeed inferior to MAP when the training dataset is very small, but the results become more similar as the size of the dataset increases.

We believe that the reason why we do not observe the poor generalization of the Bayesian models in the 1000 datapoints regime is that the low-variance directions are fairly consistent across the dataset. However, in extreme low-data cases, we cannot reliably estimate the low-variance directions leading to poor performance according to Propositions 2, 3. A detailed exploration of BNN performance in low-data regime is an exciting direction of future work.

Table 3: **Spurious correlations.** Accuracy of MAP and HMC BNNs using the MLP architecture on MNIST in low-data regime. When the dataset is very small, MAP significantly outperforms the BNN.

|          | 50 datapoints | 100 datapoints | 1000 datapoints |
|----------|---------------|----------------|-----------------|
| MAP      | 66.4%         | 74.3%          | 90.2%           |
| HMC BNN  | 53.4%         | 65.4%          | 90.3%           |

## F   Convergence of HMC accuracy with samples

In our experiments, we use 90 HMC samples from the posterior to evaluate the performance of BNNs. In this section, we verify that the Monte Carlo estimates of accuracy of the Bayesian model average converge very quickly with the number of samples, and 90 samples are sufficient for performing qualitative comparison of the methods. In Table 4 we show the accuracy for a fully-connected HMC BNN with a Gaussian prior on MNIST under different corruptions as a function of the number of samples:

Table 4: **Spurious correlations.** Accuracy of MAP and HMC BNNs using the MLP architecture on MNIST in low-data regime. When the dataset is very small, MAP significantly outperforms the BNN.

| corruption    | 10 samples | 50 samples | 100 samples | 500 samples | 1200 samples |
|---------------|------------|------------|-------------|-------------|--------------|
| MNIST         | 98.2%      | 98.19%     | 98.19%      | 98.32%      | 98.26%       |
| Impulse Noise | 85.34%     | 89.86%     | 90.68%      | 91.3%       | 91.33%       |
| Motion Blur   | 81.56%     | 81.82%     | 82.14%      | 82.47%      | 82.61%       |
| Scale         | 67.32%     | 68.69%     | 69.45%      | 69.91%      | 70.18%       |
| Brightness    | 23.66%     | 20.26%     | 22.31%      | 24.08%      | 23.4%        |
| Stripe        | 28.18%     | 30.09%     | 34.8%       | 39.26%      | 37.96%       |
| Canny Edges   | 58.79%     | 62.85%     | 63.34%      | 64.36%      | 64.32%       |

In each case, the performance estimated from 100 samples is very similar to the performance for 1200 samples. The slowest convergence is observed on the stripe corruption, but even there the performance at 100 samples is very predictive of the performance at 1200 samples.

## G   Tempered posteriors

In this section we explore the effect of posterior tempering on the performance of the MLP on MNIST. In particular, following Wenzel et al. [67] we consider the cold posteriors:

$$p_T(W|D) \propto (p(D|W)p(W))^{1/T}, \tag{4}$$

where $T \leq 1$. In Figure 8 we report the results for BNNs with Gaussian priors with variances 0.01 and 0.03 and posterior temperatures $T \in \{10^{-1}, 10^{-2}, 10^{-3}\}$. As observed by Izmailov et al. [28], lower temperatures $(10^{-2}, 10^{-3})$ improve performance under the *Gaussian noise* corruption; however, low temperatures do not help with other corruptions significantly.

## H   Proofs of the theoretical results

For convenience, in this section we assume that a constant value of 1 is appended to the input features instead of explicitly modeling a bias vector $b$. We assume that the output $f(x, W)$ of the network with parameters $W$ on an input $x$ is given by

$$f(x, W) = \psi(\phi(\ldots \phi(\phi(xW^1)W^2 + b^2))W^l + b^l), \tag{5}$$

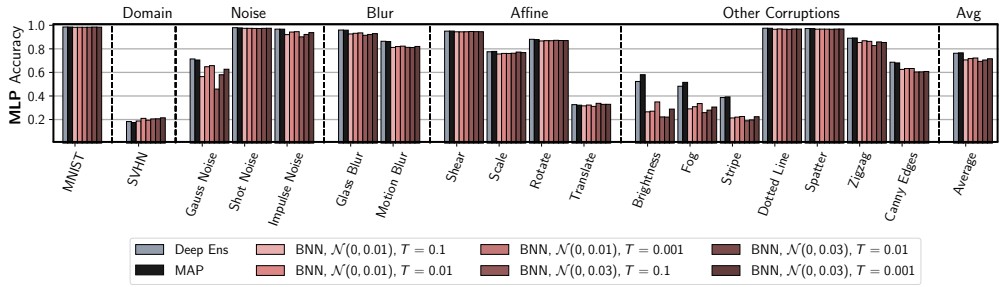

Figure 8: **Temperature ablation.** We report the performance of BNNs with Gaussian priors and tempered posteriors for different temperatures and prior scales. Low temperatures ($T = 10^{-2}$, $10^{-3}$) can provide a significant improvement on the *noise* corruptions, but do not improve the results significantly under other corruptions.

where $\phi$ are non-linearities of the intermediate layers (e.g. ReLU) and $\psi$ is the final link function (e.g. softmax).

We will also assume that the likelihood is a function $\ell(\cdot, \cdot)$ that only depends on the output of the network and the target label:

$$p(y|x, W) = \ell(y, f(x, W)).$$ (6)

For example, in classification $\ell(y, f(x, W)) = f(x, W)[y]$, the component of the output of the softmax layer corresponding to the class label $y$. Finally, we assume that the likelihood factorizes over the inputs:

$$p(\mathcal{D}|W) = \prod_{x,y \in \mathcal{D}} p(y|x, W)$$ (7)

for any collection of datapoints $\mathcal{D}$.

### H.1 Proof of Lemma 1

We restate the Lemma:

**Lemma 1** *Suppose that the input feature $x_k^i$ is equal to zero for all the examples $x_k$ in the training dataset D. Suppose the prior distribution over the parameters $p(W)$ factorizes as $p(W) = p(w_{ij}^1) \cdot p(W \setminus w_{ij}^1)$ for some neuron $j$ in the first layer, where $W \setminus w_{ij}^1$ represents all the parameters $W$ of the network except $w_{ij}^1$. Then, the posterior distribution $p(W|D)$ will also factorize and the marginal posterior over the parameter $w_{ij}^1$ will coincide with the prior:*

$$p(W|D) = p(W \setminus w_{ij}^1|D) \cdot p(w_{ij}^1).$$ (8)

*Consequently, the MAP solution will set the weight $w_{ij}^1$ to the value with maximum prior density.*

**Proof.** Let us denote the input vector $x$ without the input feature $i$ by $x^{-i}$, and the matrix $W^1$ without the row $i$ by $W_{-i}^1$. We can rewrite Equation 5 as follows:

$$f(x, W) = \psi(\phi(\dots \phi(\phi(x^{-i} W_{-i}^1 + \underbrace{x^i W_i^1}_{=0}) W^2 + b^2)) W^l + b^l).$$ (9)

As for all the training inputs $x_k$ the feature $x_k^i$ is equal to 0, the vector $x^i W_i^1$ is equal to zero and can be dropped:

$$f(x_k, W) = \psi(\phi(\dots \phi(\phi(x_k^{-i} W_{-i}^1) W^2 + b^2)) W^l + b^l)) =: f'(x_k, W_{-i}),$$ (10)

where $W_{-i}$ denotes the vector of parameters $W$ without $W_i^1$, and we defined a new function $f'$ that does not depend on $W_i$ and is equivalent to $f$ on the training data. Consequently, according to Equation 6 and Equation 7, we can write

$$p(D|W) = \prod_{k=1}^n \ell(y_k, f(x_k, W)) = \prod_{k=1}^n \ell(y_k, f'(x_k, W_{-i})).$$ (11)

In other words, the likelihood does not depend on $W_i^1$ and in particular $w_{ij}^1$ for any $j$.

Let us write down the posterior over the parameters using the factorization of the prior:

$$p(W|D) = \frac{\overbrace{p(D|W \setminus w_{ij}^1)p(W \setminus w_{ij}^1)}^{\text{does not depend on } w_{ij}^1} p(w_{ij}^1)}{Z},$$

(12)

where $Z$ is a normalizing constant that does not depend on $W$. Hence, the posterior factorizes as a product of two distributions: $p(D|W \setminus w_{ij}^1)p(W \setminus w_{ij}^1)/Z$ over $W \setminus w_{ij}^1$ and $p(w_{ij}^1)$. The marginal posterior over $w_{ij}^1$ thus coincides with the prior and is independent of the other parameters.

Maximizing the factorized posterior Equation 12 to find the MAP solution, we set the $w_{ij}^1$ to the maximum of its marginal posterior, as it is independent of the other parameters. ∎

### H.2 Formal statement and proof of Proposition 1

First, let us prove the following result for the MAP solution.

**Proposition 1'** *Consider the following assumptions:*

(a) *The input feature $x_k^i$ is equal to zero for all the examples $x_k$ in the training dataset $D$.*

(b) *The prior over the parameters factorizes as $p(W) = p(W_{-i}) \cdot p(W_i^1)$, where $W_{-i}$ is the vector of all parameters except for $W_i^1$, the row $i$ of the weight matrix $W^1$ of the first layer.*

(c) *The prior distribution $p(W_i^1)$ has maximum density at $0$.*

*Consider an input $x(c) = [x^1, \ldots, x^{i-1}, c, x^{i+1}, \ldots, x^m]$. Then, the prediction with the MAP model $W_{MAP}$ does not depend on $c$: $f(x(c), W_{MAP}) = f(x(0), W_{MAP})$.*

**Proof.** Analogous to the proof of Lemma 1, we can show that under the assumptions (a), (b) the posterior over the parameters factorizes as

$$p(W|D) = p(W^{-i}|D)p(W_i^1).$$

(13)

Then, the MAP solution will set the weights $W_i^1$ to the point of maximum density, which is $0$ under assumption (c). Consequently, based on Equation 9, we can see that the output of the MAP model will not depend on $x^i = c$. ∎

Next, we provide results for the Bayesian model average. We define *positive-homogeneous* activations as functions $\phi$ that satisfy $\phi(c \cdot x) = c \cdot \phi(x)$ for any positive scalar $c$ and any $x$. For example, ReLU and Leaky ReLU activations are positive-homogeneous.

We will call a vector $z$ of class logits (inputs to softmax) $\epsilon$-*separable* if the largest component $z_i$ is larger than all the other components by at least $\epsilon$:

$$z_i - z_j > \epsilon \quad \forall j \neq i.$$

(14)

We can prove the following general proposition.

**Proposition 1''** *We will need the following assumptions:*

(d) *The support of the prior over the parameters $W_{-i}$ is bounded: $\|W_{-i}\| < B$.*

(e) *The activations $\phi$ are positive-homogeneous and have a Lipschitz constant bounded by $L_\phi$.*

*Consider an input $x(c) = [x^1, \ldots, x^{i-1}, c, x^{i+1}, \ldots, x^m]$. Then, we can prove the following conclusions*

(2) *Suppose the link function $\psi$ is identity. Suppose also that the expectation $\mathbb{E}[\phi(\ldots \phi(\phi(W_i^1)W^2)\ldots)W^l]$ over $W$ sampled from the posterior is non-zero. Then the predictive mean under BMA (see Equation 2) on the input $x(c)$ depends on $c$.*

*(3) Suppose the link function $\psi$ is softmax. Then, for sufficiently large $c > 0$ the predicted class $\hat{y}(c) = \arg\max_y f(x(c), W)[y]$ does not depend on $x(c)$ for any sample $W$ from the posterior such that $z = \phi(\ldots\phi(\phi(W_i^1)W^2)\ldots)W^l$ is $\epsilon$-separable.*

**Proof.**  We can rewrite Equation 5 as follows:

$$
\begin{aligned}
f(x(c), W) &= \psi(\phi(\ldots\phi(\phi(x^{-i}W_{-i}^1 + cW_i^1)W^2 + b^2))W^l + b^l) = \\
&\psi(\phi(\ldots\phi(c\cdot\phi([x^{-i}W_{-i}^1]/c + W_i^1)W^2 + b^2))W^l + b^l) = \\
&\psi(c\cdot(\phi(\ldots\phi(\phi([x^{-i}W_{-i}^1]/c + W_i^1)W^2 + b^2/c))W^l + b^l/c)).
\end{aligned}
\tag{15}
$$

Now, under our assumptions the prior and hence the posterior over the weights $W_{-i}$ is bounded. As in finite-dimensional Euclidean spaces all norms are equivalent, in particular we imply that (1) the spectral norms $\|W^t\|_2 < L_W$ are bounded for all layers $t = 2, \ldots, l$ by a constant $L_W$, and (2) the Frobenious norms $\|\cdot\|$ of the bias parameters $b_t$ and the weights $W_{-i}^1$ are all bounded by a constant $B$. We will also assume that the norm of the vector $x^{-i}$ is bounded by the same constant: $\|x^{-i}\| \leq B$.

Consider the difference

$$
\begin{aligned}
&\left\| \left( \phi\left(\ldots\phi\left(\phi\left(\frac{x^{-i}W_{-i}^1}{c} + W_i^1\right)W^2 + \frac{b^2}{c}\right)\right)W^l + \frac{b^l}{c}\right) - \right. \\
&\left. \phi\left(\ldots\phi\left(\phi\left(\frac{x^{-i}W_{-i}^1}{c} + W_i^1\right)W^2 + \frac{b^2}{c}\right)\right)W^l \right\| \leq \frac{B}{c}.
\end{aligned}
\tag{16}
$$

Indeed, by the $\|b^l\| \leq B$. Next, for an arbitrary $z$ we can bound

$$
\left\| \phi\left(z + \frac{b^{l-1}}{c}\right)W^l - \phi(z)W^l \right\| \leq L_W \cdot L_\phi \cdot \frac{B}{c},
\tag{17}
$$

where we used the fact that $\phi$ is Lipschitz with $L_\phi$ and the Lipschitz constant for matrix multiplication by $W^l$ coincides with the spectral norm of $W^l$ which is bounded by $L_W$.

Using the bound in Equation 17, we have

$$
\begin{aligned}
&\left\| \left( \phi\left(\phi\left(\ldots\phi\left(\phi\left(\frac{x^{-i}W_{-i}^1}{c} + W_i^1\right)W^2 + \frac{b^2}{c}\right)W^{l-1} + \frac{b^{l-1}}{c}\right)\ldots\right)W^l + \frac{b^l}{c}\right) - \right. \\
&\left. \phi\left(\phi\left(\ldots\phi\left(\phi\left(\frac{x^{-i}W_{-i}^1}{c} + W_i^1\right)W^2 + \frac{b^2}{c}\right)\ldots\right)W^{l-1}\right)W^l \right\| \leq \frac{B}{c} + L_W \cdot L_\phi \cdot \frac{B}{c}.
\end{aligned}
\tag{18}
$$

Applying the same argument to all layers of the network (including the first layer where $\frac{x^{-i}W_{-i}^1}{c}$ plays the role analogous to $\frac{b^{l-1}}{c}$ in Equation 17), we get

$$
\begin{aligned}
&\left\| \left( \phi\left(\ldots\phi\left(\phi\left(\frac{x^{-i}W_{-i}^1}{c} + W_i^1\right)W^2 + \frac{b^2}{c}\right)\ldots\right)W^l + \frac{b^l}{c}\right) \right. \\
&\left. -\phi\left(\ldots\phi\left(\phi\left(W_i^1\right)W^2\right)\ldots\right)W^l \right\| \\
&\leq \frac{B}{c}(1 + L_W \cdot L_\phi + L_W^2 \cdot L_\phi^2 + \ldots + L_W^{l-1} \cdot L_\phi^{l-1}).
\end{aligned}
\tag{19}
$$

Choosing $c$ to be sufficiently large, we can make the bound in Equation 19 arbitrarily tight.

**Conclusion (2)**    Suppose $\psi$ is the identity. Then, we can write

$$
f(x(c), W) = c\cdot\phi\left(\ldots\phi\left(\phi\left(W_i^1\right)W^2\right)\ldots\right)W^l + \Delta,
\tag{20}
$$

where $\Delta$ is bounded: $\|\Delta\| \leq B(1 + L_W \cdot L_\phi + L_W^2 \cdot L_\phi^2 + \ldots + L_W^{l-1} \cdot L_\phi^{l-1})$. Consider the predictive mean under BMA,

$$
\mathbb{E}_W f(x(c), W) = c\cdot\underbrace{\mathbb{E}_W\phi\left(\ldots\phi\left(\phi\left(W_i^1\right)W^2\right)\ldots\right)W^l}_{\neq 0} + \underbrace{\mathbb{E}_W\Delta}_{\text{Bounded}},
\tag{21}
$$

where the first term is linear in $c$ and the second term is bounded uniformly for all $c$. Finally, we assumed that the expectation $\mathbb{E}_W \phi \left( \ldots \phi \left( \phi \left( W_i^1 \right) W^2 \right) \ldots \right) W^l \neq 0$, so for large values of $c$ the first term in Equation 21 will dominate, so the output depends on $c$.

**Conclusion (3)**    Now, consider the softmax link function $\psi$. Note that for the softmax we have $\arg\max_y \psi(c \cdot z)[y] = \arg\max_y z[y]$. In other words, multiplying the logits (inputs to the softmax) by a positive constant $c$ does not change the predicted class. So, we have

$$\hat{y}(c, W) = \arg\max_y f(x(c), W)[y] =$$

$$\arg\max_y \left( \phi \left( \ldots \phi \left( \phi \left( \frac{x^{-i} W_{-i}^1}{c} + W_i^1 \right) W^2 + \frac{b^2}{c} \right) \right) W^l + \frac{b^l}{c} \right) [y]. \tag{22}$$

Notice that $z_W = \phi(\ldots \phi(\phi(W_i^1) W^2) \ldots) W^l$ does not depend on the input $x(c)$ in any way. Furthermore, if $z_W$ is $\epsilon$-separable, with class $y_W$ corresponding to the largest component of $z_W$, then by taking

$$c > \frac{B(1 + L_W \cdot L_\phi + L_W^2 \cdot L_\phi^2 + \ldots + L_W^{l-1} \cdot L_\phi^{l-1})}{\epsilon}, \tag{23}$$

we can guarantee that the predicted class for $f(x(c), W)$ will be $y_W$ according to Equation 19. ∎

### H.3    General linear dependencies, Proposition 2

We will prove the following proposition, reducing the case of general linear dependencies to the case when an input feature is constant.

Suppose that the prior over the weights $W^1$ in the first layer is an i.i.d. Gaussian distribution $\mathcal{N}(0, \alpha^2)$, independent of the other parameters in the model. Suppose all the inputs $x_1 \ldots x_n$ in the training dataset $D$ lie in a subspace of the input space: $x_i^T c = 0$ for all $i = 1, \ldots, n$ and some constant vector $c$ such that $\sum_{i=1}^m c_i^2 = 1$.

Let us introduce a new basis $v_1, \ldots, v_m$ in the input space, such that the vector $c$ is the first basis vector. We can do so e.g. by starting with the collection of vectors $\{c, e_2, \ldots, e_m\}$, where $e_i$ are the standard basis vectors in the feature space, and using the Gram–Schmidt process to orthogonalize the vectors. We will use $V$ to denote the matrix with vectors $v_1, \ldots, v_m$ as colunms. Due to orthogonality, we have $VV^T = I$.

We can rewrite our model from Equation 5 as

$$f(x, W) = \psi(\phi(\ldots \phi(\phi(\underbrace{xV}_{\bar{x}} \underbrace{V^T W^1}_{\bar{W}^1}) W^2 + b^2)) W^l + b^l). \tag{24}$$

We can thus re-parameterize the first layer of the model by using transformed inputs $\bar{x} = xV$, and transformed weights $\bar{W}^1 = V^T W^1$. Notice that this re-parameterized model is equivalent to the original model, and doing inference in the re-parameterized model is equivalent to doing inference in the original model.

The induced prior over the weights $\bar{W}^1$ is $\mathcal{N}(0, \alpha^2 I)$, as we simply rotated the basis. Furthermore, the input $\bar{x}_k^1 = x_k^T v_1 = 0$ for all training inputs $k$. Thus, with the re-parameterized model we are in the setting of Lemma 1 and Propositions 1', 1''.

In particular, the posterior over the parameters $\bar{W}_1^1 = v_1^T W^1$ will coincide with the prior $\mathcal{N}(0, \alpha^2 I)$ (Lemma 1). The MAP solution will ignore the feature combination $\bar{x}^1 = x^T v_1$, while the BMA predictions will depend on it (Propositions 1', 1'').

### H.4    Convolutional layers, Proposition 3

Suppose that the convolutional filters in the first layer are of size $K \times K \times C$, where $C$ is the number of input channels. Let us consider the set $\hat{D}$ of size $N$ of all the patches of size $K \times K \times C$ extracted from the training images in $D$ after applying the same padding as in the first convolutional layer. Let us also denote the set of patches extracted from a fixed input image by $D_x$.

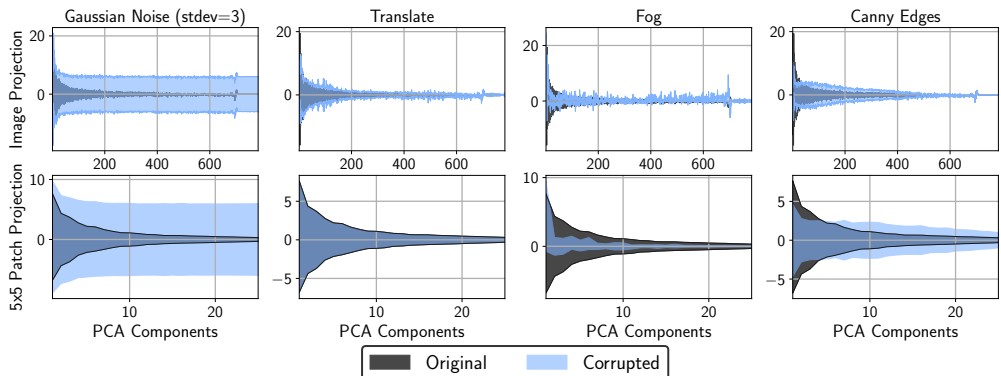

Figure 9: **Corruptions and linear dependence. Top**: The distribution (mean $\pm$ 2 std) of MNIST and MNIST-C images and **bottom**: $5 \times 5$ patches extracted from these images projected onto the corresponding principal components of the training data images and patches. *Gaussian noise* corruption breaks linear dependencies in both cases, while *Translate* does not change the projection distribution for the $5 \times 5$ patches.

A convolutional layer applied to $x$ can be thought of as a fully-connected layer applied to all patches in $D_x$ individually, and with results concatenated:

$$conv(w, x) = \left\{ \left( i, j, \sum_{a=i}^{i+k} \sum_{b=j}^{j+k} \sum_{c=1}^{C} x_{a,b,c} \cdot W^1_{a,b,c} \right) \right\}, \tag{25}$$

where $x_{a,b,c}$ is the intensity of the image at location $(a, b)$ in channel $c$, $W^1_{a,b,c}$ is the corresponding weight in the convolutional filter, and the tuples $(i, j, v)$ for all $i, j$ represent the intensities at location $(i, j)$ in the output image.

In complete analogy with Lemma 1 and Propositions 1', 1", we can show that if all the patches in the dataset $\hat{D}$ are linearly dependent, then we can re-parameterize the convolutional layer so that one of the convolutional weights will always be multiplied by 0 and will not affect the likelihood of the data. The MAP solution will set this weight to zero, while the BMA will sample this weight from the prior, and it will affect predictions.

# I    How corruptions break linear dependence in the data

In Figure 9, we visualize the projections of the original and corrupted MNIST data on the PCA components extracted from the MNIST train set and the set of all $5 \times 5$ patches of the MNIST train set. As we have seen in section 5, the former are important for the MLP robustness, while the latter are important for CNNs.

Certain corruptions increase variance along the lower PC directions more than others. For example, the *Translate* corruption does not alter the principal components of the $5 \times 5$ patches in the images, and so a convolutional BNN with a Gaussian prior is very robust to this corruption. In contrast, Gaussian noise increases variance similarly along all directions, breaking any linear dependencies present in the training data and resulting in much worse BNN performance.

# J    Analyzing other approximate inference methods

In this section, we provide additional discussion on why popular approximate inference methods SWAG and MC Dropout do not exhibit the same poor performance under covariate shift.

## J.1    Variational inference

Suppose the prior is $p(w) = \mathcal{N}(0, \alpha^2 I)$ and the variational family contains distributions of the form $q(w) = \mathcal{N}(\mu, \Lambda)$, where the mean $\mu$ and the covariance matrix $\Lambda$ are parameters. Vari-

ational inference solves the following optimization problem: maximize $\mathbb{E}_{w \sim \mathcal{N}(\mu, \Lambda)} p(D|w) - KL\left(\mathcal{N}(\mu, \Lambda) \| \mathcal{N}(0, \alpha^2 I)\right)$ with respect to $\mu, \Lambda$ [6, 35].

First, let us consider the case when the parameter $\Lambda$ is unconstrained and can be any positive-definite matrix. Suppose we are using a fully-connected network, and there exists a linear dependence in the features, as in Proposition 2. Then, there exists a direction $d$ in the parameter space of the first layer of the model, such that the projection of the weights on this direction will not affect the likelihood, and the posterior over this projection will coincide with the prior and will be independent from other directions (Proposition 2,), which is Gaussian. Consequently, the optimal variational distribution will match the prior in this projection, and will also be independent from the other directions, or, in other words, $d$ will be an eigenvector of the optimal $\Lambda$ with eigenvalue $\alpha^2$. So, variational inference with a general Gaussian variational family will suffer from the same exact issue that we identified for the true posterior. Furthermore, we can generalize this result to convolutional layers completely analogously to Proposition 3,.

Now, let us consider the mean-field variational inference (MFVI) which is commonly used in practice in Bayesian deep learning. In MFVI, the covariance matrix $\Lambda$ is constrained to be diagonal. Consequently, for general linear dependencies in the features the variational distribution will not have sufficient capacity to make the posterior over the direction $d$ independent from the other directions. As a result, MFVI will not suffer as much as exact Bayesian inference from the issue presented in Propositions 2, 3,.

One exception is the dead pixel scenario described in Section 5.1, where one of the features in the input is a constant zero. In this scenario, MFVI will have capacity to make the variational posterior over the corresponding weight match the prior, leading to the same lack of robustness described in Proposition 1 (informal).

**Empirical results.** In addition to the theoretical analysis above, we ran mean field variational inference on our fully-connected network on MNIST and evaluated robustness on the MNIST-C corruptions. Below we report the results for MFVI, MAP and HMC BNN with a Gaussian prior:

Table 5: Accuracy of MAP, MFVI and HMC BNNs under different corruptions. MFVI is more robust than HMC and even outperforms the MAP solution for some of the corruptions.

| method | CIFAR-10 | Gaussian Noise | Motion Blur | Scale | Brightness | Stripe | Canny Edges |
|--------|----------|----------------|-------------|-------|------------|--------|-------------|
| MAP | **98.5%** | **70.6%** | **86.7%** | **77.6%** | 50.6% | 34.1% | 68% |
| MFVI | 97.9% | 62.5% | 82.2% | 70.5% | **68.9%** | **47.7%** | **70%** |
| HMC | 98.2% | 43.2% | 82.1% | 69.5% | 22.3% | 34.8% | 63.3% |

As expected from our theoretical analysis, MFVI is much more robust to noise than HMC BNNs. However, on some of the corruptions (Gaussian Noise, Motion Blur, Scale) MFVI underperforms the MAP solution. At the same time, MFVI even outperforms MAP on Brightness, Stripe and Canny Edges.

## J.2 SWAG

SWA-Gaussian (SWAG) [43] approximates the posterior distribution as a multivariate Gaussian with the SWA solution [27] as its mean. To construct the covariance matrix of this posterior, either the second moment (SWAG-Diagonal) or the sample covariance matrix of the SGD iterates is used. For any linear dependencies in the training data, the corresponding combinations of weights become closer to zero in later SGD iterates due to weight decay. Since SWAG only uses the last $K$ iterates in constructing its posterior, the resulting posterior will likely have very low variance in the directions of any linear dependencies. Furthermore, because SGD is often initialized at low magnitude weights, even the earlier iterates will likely have weights close to zero in these directions.

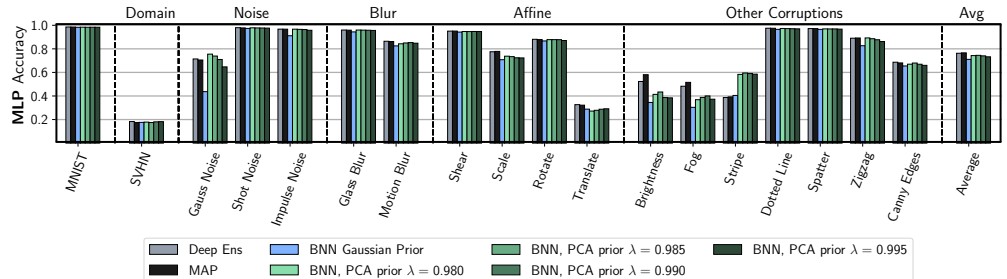

Figure 10: **General PCA priors.** Performance of the PCA priors introduced in Appendix K for various decay rates $\lambda$. PCA priors generally improve performance significantly under *Gaussian noise* and *Stripe*. Lower decay rates $\lambda$ provide better results under *Gaussian noise*.

### J.3   MC Dropout

MC Dropout applies dropout at both train and test time, thus allowing computation of model uncertainty from a single network by treating stochastic forward passes through the network as posterior samples. The full model learned at train time is still an approximate MAP solution, and thus will be minimally affected by linear dependencies in the data being broken at test time. As for the test-time dropout, we can conclude that if the expected output of the network is not affected by linear dependencies being broken, then any subset of that network (containing a subset of the network's hidden units) would be similarly unaffected. Additionally, if dropping an input breaks a linear dependency from the training data, the network (as an approximate MAP solution) is robust to such a shift.

## K   General PCA Priors

In section 6 we introduced the *EmpCov* prior, which improves robustness to covariate shift by aligning with the training dataset's principal components. Following the notation used in section 6, we can define a more general family of PCA priors as

$$p(w^1) = \mathcal{N}(0, \alpha V \text{diag}(s) V^T + \epsilon I), \quad s_i = f(i) \tag{26}$$

where for an architecture with $n_w$ first layer weights, $s$ is a length $n_w$ vector, $\text{diag}(s)$ is the $n_w \times n_w$ diagonal matrix with $s$ as its diagonal, and $V$ is an $n_w \times n_w$ matrix such that the $i^{th}$ column of $V$ is the $i^{th}$ eigenvector of $\Sigma$.

The *EmpCov* prior is the PCA prior where $f(i)$ returns the $i^{th}$ eigenvalue (explained variance) of $\Sigma$. However, there might be cases where we do not want to directly use the empirical covariance, and instead use an alternate $f$. For example, in a dataset of digits written on a variety of different wallpapers, the eigenvalues for principal components corresponding to the wallpaper pattern could be much higher than those corresponding to the digit. If the task is to identify the digit, using *EmpCov* might be too restrictive on digit-related features relative to wallpaper-related features.

We examine alternative PCA priors where $f(i) = \lambda^i$ for different decay rates $\lambda$. We evaluate BNNs with these priors on MNIST-C, and find that the choice of decay rate can significantly alter the performance on various corruptions. Using priors with faster decay rates (smaller $\lambda$) can provide noticeable improvement on Gaussian Noise and Zigzag corruptions, while the opposite occurs in corruptions like Translate and Fog. Connecting this result back to Appendix I and Figure 9, we see that the corruptions where faster decay rates improve performance are often the ones which add more noise along the smallest principal components.

## L   Effect of non-zero mean corruptions

In Figure 11, we report the results of deep ensembles, MAP and BNNs with Gaussian and *EmpCov* priors under various corruptions using the CNN architecture on MNIST. *EmpCov* improves perfor-

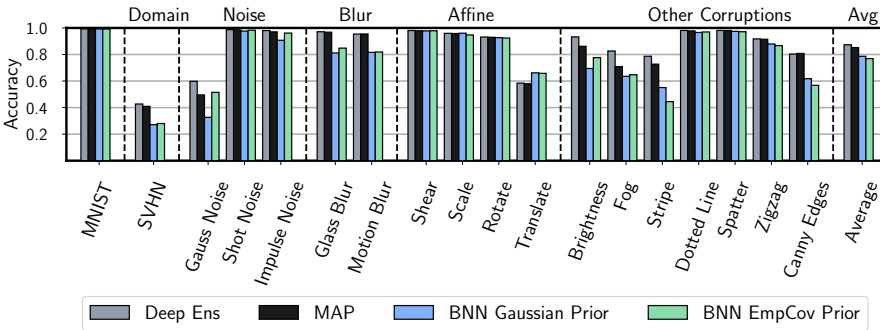

Figure 11: **MNIST CNN results.** Test accuracy under covariate shift for deep ensembles, MAP optimization with SGD, and BNN with Gaussian and *EmpCov* priors.

mance on all of the noise corruptions. For example, on the *Gaussian noise* corruption, *EmpCov* achieves 58.3% accuracy while the Gaussian prior achieves 32.7% accuracy; similarly on the *impulse noise* the results are 96.5% and 90.73% respectively.

However, *EmpCov* does not improve the results significantly on *brightness* or *fog*, and even hurts the performance slightly on *stripe*. Below, we explain that these corruptions are non-zero mean, and the performance is affected by the sum of the filter weights. We thus propose the SumFilter prior which greatly improves the performance on these corruptions.

### L.1 Non-zero mean corruptions

As we have seen in various experiments (e.g. Figure 2, Figure 7), convolutional Bayesian neural networks are particularly susceptible to the *brightness* and *fog* corruptions on MNIST-C. Both of these corruptions are not zero-mean: they shift the average value of the input features by 1.44 and 0.89 standard deviations respectively. In order to understand why non-zero mean corruptions can be problematic, let us consider a simplified corruption that applies a constant shift $c$ to all the pixels in the input image. Ignoring the boundary effects, the convolutional layers are linear in their input. Denoting the output of the convolution with a filter $w$ on an input $x$ as $conv(w, x)$, and an image with all pixels equal to 1 as $\mathbb{1}$ we can write

$$conv(w, x + c \cdot \mathbb{1}) = conv(w, x) + c \cdot conv(w, \mathbb{1}) = conv(w, x) + \mathbb{1} \cdot c \cdot \sum_{a,b} w_{a,b}, \quad (27)$$

where the last term represents an image of the same size as the output of the $conv(\cdot, \cdot)$ but with all pixels equal to the sum of the weights in the convolutional filter $w$ multiplied by $c$. So, if the input of the convolution is shifted by a constant value $c$, the output will be shifted by a constant value $c \cdot \sum_{a,b} w_{a,b}$.

As the convolutional layer is typically followed by an activation such as ReLU, the shift in the output of the convolution can significantly hinder the performance of the network. For example, suppose $c \cdot \sum_{a,b} w_{a,b}$ is a negative value such that all the output pixels in $conv(w, x) + \mathbb{1} \cdot c \cdot \sum_{a,b} w_{a,b}$ are negative. In this case, the output of the ReLU activation applied after the convolutional filter will be 0 at all output locations, making it impossible to use the learned features to make predictions.

In the next section, we propose a prior that reduces the sum $\sum_{a,b} w_{a,b}$ of the filter weights, and show that it significantly improves robustness to multiple corruptions, including *fog* and *brightness*.

### L.2 SumFilter prior

As we've discussed, if the sum of filter weights for CNNs is zero, then corrupting the input by adding a constant has no effect on our predictions. We use this insight to propose a novel prior that constrains

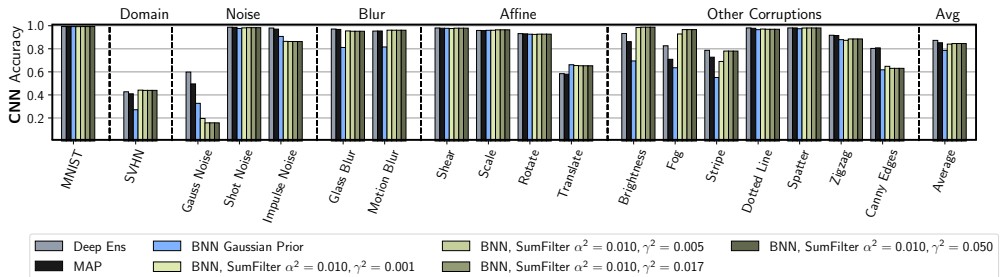

Figure 12: **SumFilter priors.** Performance of BNNs with the *SumFilter* priors introduced in subsection L.2 for the CNN architecture on MNIST. *SumFilter* priors do not improve the performance under *Gaussian noise* unlike *EmpCov priors*, but provide a significant improvement on the *Brightness* and *Fog* corruptions.

the sum of the filter weights. More specifically, we place a Gaussian prior on the parameters and Laplace prior on the sum of the weights:

$$p(w) \sim \mathcal{N}\left(w|0, \alpha^2 I\right) \times \text{Laplace}\Big(\sum_{\text{filter}} w|0, \gamma^2\Big). \tag{28}$$

For our experiments, we only place the additional Laplace prior on the sum of weights in first layer filters. An alternative version could place the prior over filter sums in subsequent layers, which may be useful for deeper networks.

### L.3  Experiments

Figure 12 shows that this prior substantially improves the performance of a convolutional BNN on MNIST-C. The BNN with a filter sum prior yields a better or comparable performance to MAP for all MNIST corruptions, with the exception of *Canny Edges* and *Impulse Noise*. We also implemented this prior for MLPs, but found that it only improved BNN performance on two corruptions, fog and brightness. Overall, this prior addresses a more specific issue than *EmpCov*, and we would not expect it to be applicable to as many forms of covariate shift.

## M    Example: Bayesian NALU under covariate shift

The Neural Arithmetic Logic Unit (NALU) [66] is an architecture which can learn arithmetic functions that extrapolate to values outside those observed during training. A portion of the unit is of the form $\prod_{j=1}^{m} |x_j|^{w_j}$, and in this section we examine a simplified form of this unit in order to demonstrate an instance where nonlinear dependencies hurt BMA under covariate shift.

Let's consider the NALU-inspired architecture with input features $x^1, \ldots, x^m$ that takes the form $f(x, w) = \prod_{j=1}^{m}(x^j)^{w_j}$. Suppose the prior over the weights $w = [w_1, \ldots, w_m]$ is an i.i.d. Gaussian distribution $\mathcal{N}(0, \alpha^2)$. Suppose all inputs $x_1, \ldots x_n$ in training dataset $\mathcal{D}$ lie in a subspace of the input space: $\prod_{j=1}^{m}(x_i^j)^{p_j} = 1$ for all $i = 1, \ldots, n$ and some constant vector $p$ such that $\sum_{j=1}^{m} p_j^2 = 1$. Following the same approach as subsection H.3, we can introduce a new basis $v_1, \ldots, v_m$ in the input space such that $v_1 = p$. We can similarly re-parameterize the model using the weights rotated into this new basis, $\bar{w} = w^T v_1, \ldots, w^T v_m$, and it follows that $w_i = \bar{w}_1 \cdot v_1^i + \cdots + \bar{w}_m \cdot v_m^i$ for all $i = 1, \ldots, n$. Using the corresponding transformed inputs $\bar{x}^i = \prod_{j=1}^{m}(x^j)^{v_i^j}$ for all $i = 1, \ldots, n$, we can rewrite our model as follows:

$$f(x, w) = f(\bar{x}, \bar{w}) = \big(\prod_{j=2}^{m}(\bar{x}^j)^{\bar{w}_j}\big) \cdot \underbrace{(\bar{x}^1)^{\bar{w}_1}}_{=1}. \tag{29}$$

Since $f(\bar{x}, \bar{w})$ does not depend on $\bar{w}_1$ for all $\bar{x} \in \bar{\mathcal{D}}$, we can follow the same reasoning from subsection H.1 to conclude that the marginal posterior over $\bar{w}_1$ coincides with the induced prior. Since $\bar{w}$ is the result of simply rotating $w$ into a new basis, it also follows that the induced prior over $\bar{w}$ is $\mathcal{N}(0, \alpha^2 I)$, and that the posterior can be factorized as $p(\bar{w}|\mathcal{D}) = p(\bar{w} \setminus \bar{w}_1|\mathcal{D}) \cdot p(\bar{w}_1)$.

Consider a test input $\bar{x}_k(c) = [c, \bar{x}_k^2, \ldots, \bar{x}_k^m]$. The predictive mean under BMA will be:

$$\mathbb{E}_{\bar{w}} f(\bar{x}_k(c), \bar{w}) = \mathbb{E}_{\bar{w} \setminus \bar{w}_1} \prod_{j=2}^{m} (\bar{x}_k^j)^{\bar{w}_j} \cdot \mathbb{E}_{\bar{w}_1} c^{\bar{w}_1}. \tag{30}$$

Thus the predictive mean depends upon $c$, and so the BMA will not be robust to the nonlinear dependency being broken at test time. In comparison, the MAP solution would set $\bar{w}_1 = 0$, and its predictions would not be affected by $c$.

While the dependency described in this section may not necessarily be common in real datasets, we highlight this example to demonstrate how a nonlinear dependency can still hurt BMA robustness. This further demonstrates how the BMA issue we've identified does not only involve *linear* dependencies, but rather involves dependencies which have some relationship to the model architecture.

## N    Dead neurons

Neural network models can often contain *dead neurons*: hidden units which output zero for all inputs in the training set. This behaviour occurs in classical training when a neuron is knocked off the training data manifold, resulting in zero non-regularized gradients for the corresponding weights and thus an inability to train the neuron using the gradient signal from the non-regularized loss. However, we can envision scenarios where a significant portion of the BNN posterior distribution contains models with dead neurons, such as when using very deep, overparameterized architectures.

Let us consider the posterior distribution over the parameters $W$ conditioned on the parameters $W^1, W^2, b^2, \ldots, W^k, b^k$ of the first $k$ layers, where we use the notation of Appendix H. Suppose for the parameters $W^1, W^2, b^2, \ldots, W^k, b^k$ the $k$-th layer contains a dead neuron, i.e. an output that is $0$ for all the inputs $x_j$ in the training dataset $D$. Then, consider the sub-network containing layers $k+1, \ldots, l$. For this sub-network, the output of a dead neuron in the $k$-th layer is an input that is $0$ for all training inputs. We can then apply the same reasoning as we did in subsection H.1, subsection H.2 to show that there will exist a direction in the parameters $W^k$ of the $k + 1$-st layer, such that along this direction the posterior *conditioned* on the parameters $W^1, W^2, b^2, \ldots, W^k, b^k$ coincides with the prior (under the assumption that the prior over the parameters $W^k$ is iid and independent of the other parameters). If a test input is corrupted in a way that activates the dead neuron, the predictive distribution of the BMA *conditioned* on the parameters $W^1, W^2, b^2, \ldots, W^k, b^k$ will change.

## O    Bayesian linear regression under covariate shift

We examine the case of Bayesian linear regression under covariate shift. Let us define the following Bayesian linear regression model:

$$y = w^\top \phi(x, z) + \epsilon(x) \tag{31}$$

$$\epsilon \sim \mathcal{N}(0, \sigma^2) \tag{32}$$

where $w \in \mathbb{R}^d$ are linear weights and $z$ are the deterministic parameters of the basis function $\phi$. We consider the dataset $D = \{(x_i, y_i)\}_{i=1}^{n}$ and define $y := (y_1, \ldots, y_N)^\top$, $X := (x_1, \ldots, x_n)^\top$, and $\Phi := (\phi(x_1, z), \ldots, \phi(x_n, z))^\top$

The likelihood function is given by:

$$p(y|X, w, \sigma^2) = \prod_{i=1}^{n} \mathcal{N}(y_i | w^\top \phi(x_i, z), \sigma^2). \tag{33}$$

Let us choose a conjugate prior on the weights:

$$p(w) = \mathcal{N}(w|\mu_0, \Sigma_0), \tag{34}$$

The posterior distribution is given by:

$$p(w|\mathcal{D}) \propto \mathcal{N}(w|\mu_0, \Sigma_0) \times \prod_{i=1}^{n} \mathcal{N}(y_i|w^\top \phi(x_i, z), \sigma^2)$$
$$= \mathcal{N}(w|\mu, \Sigma),$$
$$\mu = \Sigma \left( \Sigma_0^{-1} \mu_0 + \frac{1}{\sigma^2} \Phi^\top y \right),$$
$$\Sigma^{-1} = \Sigma_0^{-1} + \frac{1}{\sigma^2} \Phi^\top \Phi.$$

The MAP solution is therefore equal to the mean,

$$w_{\text{MAP}} = \Sigma \left( \Sigma_0^{-1} \mu_0 + \frac{1}{\sigma^2} \Phi^\top y \right) = \left( \Sigma_0^{-1} + \frac{1}{\sigma^2} \Phi^\top \Phi \right)^{-1} \left( \Sigma_0^{-1} \mu_0 + \frac{1}{\sigma^2} \Phi^\top y \right) \tag{35}$$

Thus, we see that the BMA and MAP predictions coincide in Bayesian linear regression, and both will have equivalent performance under covariate shift in terms of accuracy.

**What happens away from the data distribution?** If the data distribution spans the entire input space, than the posterior will contract in every direction in the weight space. However, if the data lies in a linear (or affine, if we are using a Gaussian prior) subspace of the input space, there will be directions in the parameter space for which the posterior would coincide with the prior. Now, if a test input does not lie in the same subspace, the predictions on that input would be affected by the shift vector according to the prior. Specifically, if the input $x$ is shifted from the subspace containing the data by a vector $v$ orthogonal to the subspace, then the predictions between $x$ and its projection to the subspace would differ by $w^T v$, where $w \sim \mathcal{N}(\mu_0, \Sigma_0)$, which is itself $\mathcal{N}(\mu_0^T v, v^T \Sigma_0 v)$. Assuming the prior is zero-mean, the mean of the prediction would not be affected by the shift, but the uncertainty will be highly affected. The MAP solution on the other hand does not model uncertainty.

# P An optimization perspective on the covariate shift problem.

In this section, we examine SGD's robustness to covariate shift from an optimization perspective.

## P.1 Effect of regularization and initialization on SGD's robustness to covariate shift

In Section N, we discussed how SGD pushes the weights that correspond to dead neurons, a generalization of the dead pixels analysis, towards zero thanks to the regularization term. In this section, we study the effect of regularization and initialization on SGD's robustness under covariate shift.

**Regularization**
To study the effect of regularization on SGD's robustness under covarite shift, we hold all hyperparameters fixed and we change the value of the regularization parameter. Figure 13 shows the outcome. Most initialization schemes for neural networks initialize the weights with values close to zero, hence we expect SGD not perform as poorly on out-of-distribution data as HMC on these networks even without regularization. Therefore, we see that SGD without regularization (reg = 0.0) is still competitive with reasonably regularized SGD.

**Initialization**
The default initialization scheme for fully-connected layers in Pytorch for example is the He initialization [24]. We use a uniform initialization $\mathcal{U}(-b, b)$ and study the effect of varying $b$ on the performance of SGD under covarite shift for a fully-connected neural network. Figure 14 shows our empirical results, where smaller weights result in better generalization on most of the corruptions.

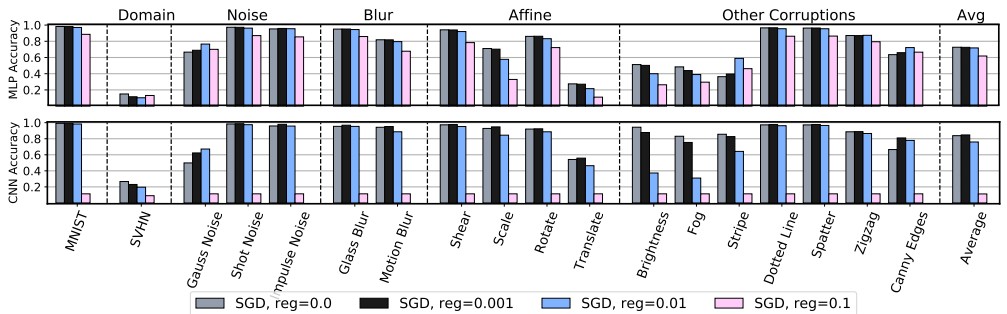

Figure 13: **Effect of regularization on SGD's performance on corrupted MNIST.** Accuracy for the following values of the regularization parameter: 0.0, 0.001, 0.01, and 0.1. **Top**: Fully-connected network; **bottom**: Convolutional neural network. Regularization helps improve the performance on some corruptions, such as Gaussian noise, but its absence does not affect SGD's robustness under covariate shift because the weights are initialized at small values.

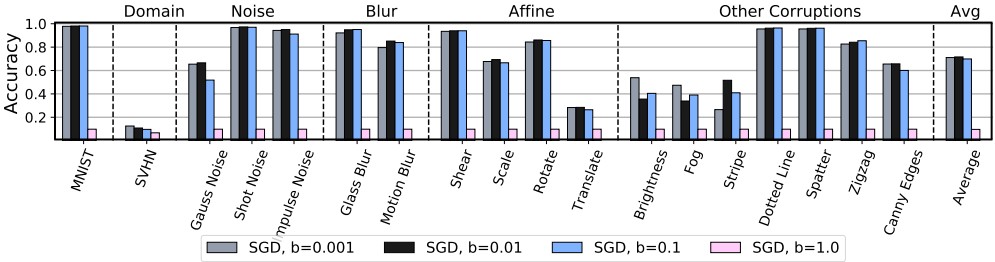

Figure 14: **Effect of initialization on SGD's performance on corrupted MNIST with MLP.** The weights are initialized using a uniform distribution $\mathcal{U}(-b, b)$, and we consider the following values for $b$: 0.001, 0.01, 0.1, and 1.0. All experiments were run without regularization. For most corruptions, initializing the weights at smaller values leads to better robustness to covariate shift.

## P.2 Other stochastic optimizers

In addition to SGD, we examine the performance of Adam [34], Adadelta [70], L-BFGS [55, 41] on corrupted MNIST. Figure 15 shows the results for all 4 algorithms on the MNIST dataset under covariate shift, for both fully-connected and convolutional neural networks. We see that SGD, Adam and Adadelta have comparable performance for convolutional neural networks, whereas SGD has an edge over both algorithms on MLP. L-BFGS provides a comparatively poor performance and we hypothesise that it is due to the lack of regularization. Naive regularization of the objective function does not improve the performance of L-BFGS.

## P.3 Loss surface analysis

There have been several works that tried to characterize the geometric properties of the loss landscape and describe its connection to the generalization performance of neural networks. In particular, it is widely believed that flat minima are able to provide better generalization [26, 32]. Intuitively, the test distribution introduces a horizontal shift in the loss landscape which makes minima that lie in flat regions of the loss surface perform well for both train and test datasets. From the other side, it is well-known that SGD produces flat minima. Hence, we would like to understand the type of distortions that corruptions in the corrupted CIFAR-10 dataset introduce in the loss surface, and evaluate the potential advantage of flat minima in this context.

In the same fashion as Li et al. [40], we visualize the effect of the Gaussian noise corruption on the loss surface for different intensity levels as shown in Figure 16. These plots are produced for two random directions of the parameter space for a ResNet-56 network. We observe that high levels of intensity make the loss surface more flat, but result in a worse test loss overall. We can see visually that the mode in the central flat region, that we denote $w_0$, is less affected by the corruption than a

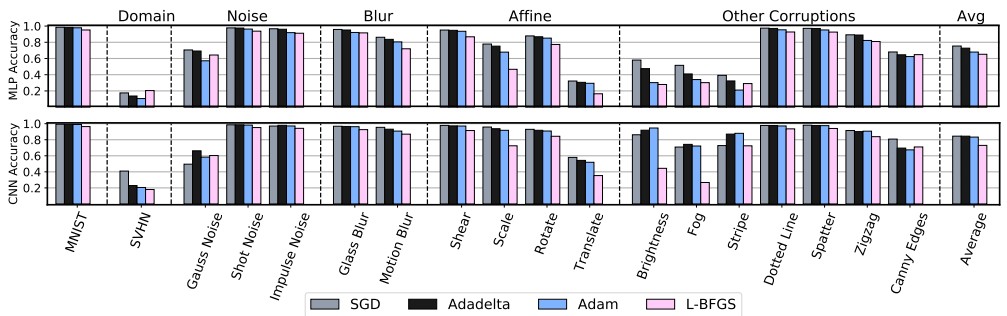

Figure 15: **Robustness on MNIST for different stochastic optimizers.** Accuracy for SGD, Adadelta, Adam and L-BFGS on MNIST under covariate shift. **Top**: Fully-connected network; **bottom**: Convolutional neural network. Adam and Adadelta provide competitive performance with SGD for most corruptions. However, SGD is better on the MLP architecture for some corruptions whereas Adam and Adadelta are better on the *same* corruptions with the CNN architecture.

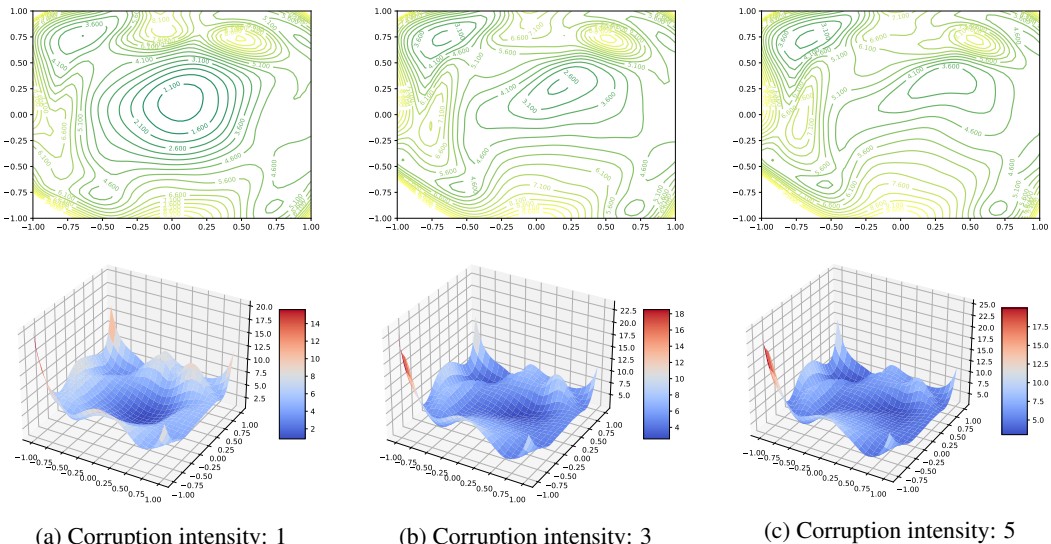

(a) Corruption intensity: 1      (b) Corruption intensity: 3      (c) Corruption intensity: 5

Figure 16: 2D (**top**) and 3D **bottom** visualizations of the loss surface of a ResNet-56 network on corrupted CIFAR-10 with different intensity levels of the Gaussian noise corruption.

solution picked at random. Figure 17 shows the loss difference between different solutions including $w_0$, that we call *optimal*, and the new mode for each corruption intensity. We can see that the mode is indeed less affected by the corruptions than other randomly selected solutions of the same loss region.

## Q   Licensing

The MNIST dataset is made available under the terms of the Creative Commons Attribution-Share Alike 3.0 license. The CIFAR-10 dataset is made available under the MIT license. Our code is a fork of the Google Research repository at https://github.com/google-research/google-research, which has source files released under the Apache 2.0 license.

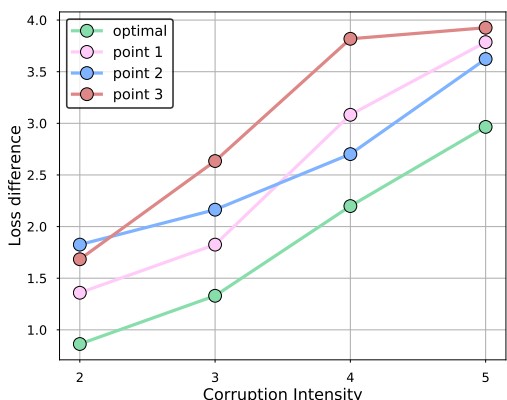

Figure 17: Loss difference between different random solutions, including the mode found through standard SGD training (*optimal* in the legend), and the new mode for each corruption intensity.