# OpenReview forum: "Dangers of Bayesian Model Averaging under Covariate Shift"
_NeurIPS.cc/2021/Conference — NeurIPS 2021 Poster_

### Official Review · Reviewer_SBkb · 2021-07-06

**Rating:** 6
**Confidence:** 4

**Summary:**

This paper begins with the observation that BMA underperforms MAP or ensembling under distribution shift and corruptions.
It observes that this could well be because of weights in the first layer reverting to the prior along dimensions which are off the data-manifold.
These weights can take on high values, even though their mean is low, and the resulting influence on the prediction might be high because of the nonlinearity.
The authors empirically support their analysis by proposing a novel data-dependent prior which has low variance along these dimensions.
They find that in some settings this prior improves accuracy under corruption, although perhaps not always.

**Limitations And Societal Impact:**

This is fine on the whole. More acknowledgement in the introduction of the fact that the prior is data-dependent and the limitations this entails, as well as the fact that they seem to work slightly inconsistently, might help slightly.

**Main Review:**

# Overall review
There were parts of this paper that I thought were really excellent.
The analysis in section 5 is very strong, and I particularly enjoyed the experiment in Figure 4, which is a clever test of the hypothesis.

I am not currently recommending acceptance, although I am uncertain and borderline.
I felt that there might be some alternative explanations for the benefit of ensembling/MAP over BMA which were not considered or properly ruled out (see below).
At the same time, the main empirical justification for your diagnosis of the problem lies in the effectiveness of the EmpCov prior, but you noted that the prior was not effective in MNIST for CNNs, although I couldn't find any figures showing this result.
Off the back of the strength of the analysis of section 5, I believe that you have indeed identified *an* issue for BMA under distribution shift, but I don't yet feel confident that this is the main reason for the performance gap that you discuss.

## Originality
On the whole, the work makes some novel contributions.
However, large parts of the text and figures build heavily off of Izmailov et al. "What are Bayesian Neural Network Posteriors Really Like?". ICML. 2021. It isn't really until page 5 that things really start to take off with novel work.

## Quality
The quality was generally high, and I was impressed by the number of ablations performed.
I make a note below about the terminology "covariate shift", which I think may not be correct here.
There were also some textual descriptions that referred to figures that I could not find.

## Clarity
On the whole, I felt that the text and figures were clear.

## Significance
I could not quite tell how significant the work seems likely to be.
The authors suggest in the introduction that their novel prior could be generally adopted, but don't heavily discuss the risks and limitations of data-dependent priors.
Setting aside the prior, it seems likely that the work could motivate further use of MAP and ensembles, however since these are already heavily used it is unclear how much this impact will extend beyond what was already known in prior work.
The analysis of S5 feels quite satisfying though, and perhaps it will unlock deeper insights in the future.

# Major comments

### Suppose the HMC is working, is your MC integration working?

I'll grant the assumption HMC provides a good set of samples from the true posterior p(w|D) (though reasonable people could disagree about this assumption).
But it doesn't necessarily follow that the MC integral (your Equation 2) using HMC samples is a good estimate of the marginal p(y|x).
An integration failure seems especially likely given the comparatively small number of samples used for integration (90) compared to the dimensionality of the space.
Would you be able to validate the accuracy of your integration?

I understand that the non-linearity means that even under perfect integration the expectation will not match the MAP result.
It would be nice to demonstrate directly that this noise is *actually* amplified by the non-linearity, rather than being cancelled out higher up the network.
I wouldn't be surprised if in large and deep networks this effect is somewhat corrected for naturally (for variational inference, at least, it has been found to be somewhat corrected by 'over-pruning' early weights that are set near the prior (see Trippe and Turner "Overpruning in Variational Bayesian Neural Networks". 2018).
The question is how big the effect you've identified is compared to orthers.
I understand you are computationally constrained from getting much bigger models with HMC, of course.

### Comparison with Domingos, P. "Bayesian averaging of classifiers and the overfitting problem." ICML 2000.

He made a similar observation to yours: empirical failure of principled BMA compare to the more naive ensembling which is not weighted by p(w|D).
But his work was in the context of trees, rather than NNs with HMC.
His explanation of the failure of BMA, is that the BMA overfits to parameter-values that have particularly high probability under the data.
HMC won't have the same overfitting problem, but I think your analysis might point to something similar going on based on the fact that the training dataset is only a manifold of the overall-dataspace.
I don't think this is inconsistent with your analysis, and it might be either a special case or competing explanation, but seems relevant to your S5.4.
(Note some commentary in Minka TP. "Bayesian Model Averaging Is Not Model Combination". 2000. which may also shed some light on the issues.)

### Comparison with Izmailov et al. "What are Bayesian Neural Network Posteriors Really Like?". ICML. 2021.

It seems like there's a reasonable amount of overlap between much of this work and results in this paper by Izmailov et al.
In particular, it isn't until section 5 that any substantially new results or observations appear to be introduced.
The main differences before that point (as far as I can tell) are the addition of MNIST and the use of smaller LeNet-style networks rather than ResNet-20 (presumably for computational reasons).
Figure 1(b,c) is also completely novel, but I would have actually liked more investigation of this interesting figure!
This is not necessarily a huge issue---the analysis and discussion of the results and observations extends beyond the other work---but I wonder if the paper would be stronger if more space was devoted to new results.

### Figure 4 and S5.2 are really excellent
This is such a clever experiment and a great result.
It should be more heavily emphasised earlier on.

### Use of the term 'covariate shift'
ln 15: strictly speaking, 'covariate shift' is a special case of the target distribution differing from the training distribution, as you note in S2.
The 'OOD generalization tasks' you refer to not only change p(x) but also change p(y|x), and are therefore not covariate shift.
I think the CIFAR-10-C datasets are probably also not covariate shift.
For example, if you take some dataset X and add Gaussian noise to X such that $X' = X + \epsilon$, but keep $\mathbf{y}$ constant, then it seems unlikely that p(y|x) is unchanged.
After all, there are presumably images in the original (un-noised) data distribution which could only have been from class 1 but which are now ambiguous between class 1/2.
I'm not certain that this undermines anything you have specifically introduced, but it means that some of the literature you relate to or draw on might need to be interpreted more carefully and that results that are specific to covariate shift might not actually apply in the settings which you are exploring.

Sentences like ln 116 "In this case, there is no semantic distribution shift:" I'm not sure if this is true given the above.
Or something like MNIST to SVHN, it isn't clear what p(y|x) in MNIST is for SVHN images---MNIST can be perfectly labelled with a p(y|x) that has zero accuracy on crop/greyed SVHN and vice versa.
To be clear, I don't really think that this point is all that important to the contribution of your paper---and it hasn't formed a significant part of my score.

### EmpCov in CNN on MNIST
>In Appendix I, we provide a detailed analysis of the performance of the CNN architecture on MNIST.
Surprisingly, we found that using the EmpCov prior by itself does not provide a large improvement in this case.

I was expecting to find a chart of EmpCov on CNNs in MNIST in appendix I and I did not, did I miss it?
I would perhaps have expected to find it in Figure 7 or 11, but it seemed not to be there.
When you say in line 311 it "does not provide a large improvement", does it provide any improvement? Does it hurt?
If so, why?

In line 312 you write that the issue is "specific to this particular setting", and in the appendix you explain that the "brightness" and "fog" corruptions are especially problematic.
But I don't immediately see why this would be much worse for MNIST than CIFAR, it seems worth substantiating that a little.
Presumably with a MNIST CNN equivalent of Figure 7 we could see if there was a similar effect for things other than "brightness" and "fog".
If I understand it right, what you've written is consistent with there being a benefit to EmpCov for MNIST CNNs for all the corruptions except those two.
Is that the case?

Given that the proposal of a new prior is both a major contribution of your paper and one of the main empirical evaluations that the issue you have identified is a major issue for the difference between BMA and ensembling, I would definitely devote more space to examining these issues and the SumFilter at the expense of some of the repetition of results from prior work.

# Minor remarks
ln 39: "similar to a draw from the Gaussian prior."
It's not clear what sense of 'similar' you are referring to, and you should probably substantiate it.
Do you just mean it is noisy?
How much like a draw from a Gaussian is it?
In what way?
What do the means of the model average look like for these weights? (i.e., it would be interesting to contrast this plot with (d) BNN weights averaged over N samples and see if it looks like (c))

ln 298: I'd like a little more discussion of the use of this as a prior.
In particular, it is likely to lead to weight posteriors which are highly overconfident in some parts of the space.
If this were used for any kind of sequential inference with multiple datasets, say, it would be quite inappropriate.
That's all fine, it's just worth noting.

**Time Spent Reviewing:**

3

---

> ### Author Response · Authors · 2021-08-10
> **Response to Reviewer SBkb**
>
> Thank you for your thoughtful and detailed review! We are happy that you found many aspects of the paper to be strong, and we aim to address your concerns below. In particular, we have conducted an experiment, showing that the Monte Carlo estimates of accuracy of the Bayesian model average converge very quickly with the number of samples, and the 90 samples are sufficient for performing qualitative comparison of the methods. In the table below we show the accuracy for a fully-connected HMC BNN with a Gaussian prior on MNIST under different corruptions as a function of the number of samples:
>
> | corruption    |   10 samples |   50 samples |   100 samples |   500 samples |   1200 samples |
> |:--------------|-------------:|-------------:|--------------:|--------------:|---------------:|
> | MNIST      |       0.982  |       0.9819 |        0.9819 |        0.9832 |         0.9826 |
> | impulse_noise |       0.8534 |       0.8986 |        0.9068 |        0.913  |         0.9133 |
> | motion_blur   |       0.8156 |       0.8182 |        0.8214 |        0.8247 |         0.8261 |
> | scale         |       0.6732 |       0.6869 |        0.6945 |        0.6991 |         0.7018 |
> | brightness    |       0.2366 |       0.2026 |        0.2231 |        0.2408 |         0.234  |
> | stripe        |       0.2818 |       0.3009 |        0.348  |        0.3926 |         0.3796 |
> | canny_edges   |       0.5879 |       0.6285 |        0.6334 |        0.6436 |         0.6432 |
>
> In each case, the performance estimated from 100 samples is very similar to the performance for 1200 samples. The slowest convergence  is observed on the stripe corruption, but even there the performance at 100 samples is very predictive of the performance at 1200 samples.
>
>
> While there may be several explanations for the poor generalization of BNNs under covariate shift, we believe that the insights that we provide, alongside a new prior which significantly improves robustness and was directly inspired by these insights, are important contributions in Bayesian deep learning. Indeed, this issue is crucial to the applicability of Bayesian neural networks in the real world, where training and test distributions are often slightly different, but we still want to make useful predictions. Moreover, we believe that our empirical  analysis in Section 5 and the results for the *EmpCov* prior (including CNN on MNIST, please see below) show that the mechanisms that we identify play an important role in the poor generalization of BNNs under covariate shift. We also note that we do consider complementary hypotheses, such as the flatness of solutions, in the appendix, but we focus on what we believe to be the most direct and actionable explanation.
>
> We hope you can consider our responses and clarifications, as well as the timeliness and practical importance of these contributions, in your final assessment.
>
> ### Novelty
>
> While it was briefly observed that Bayesian neural networks generalize poorly under covariate shift in [1], our paper makes many novel contributions inspired by this observation. In [1], the poor OOD generalization was only briefly presented as a surprising phenomenon without an explanation. In this paper, we (1) show that the phenomenon holds quite generally for a range of priors and several new problems; (2) we show that the poor OOD generalization is not resolved by posterior tempering, in contradiction to the hypothesis presented in [1]; (3) provide both a theoretical and empirical analysis in Section 5, which forms a key part of our contributions; (4) finally, we propose *EmpCov*, a prior directly based on our analysis in Section 5, which we show improves robustness in a variety of settings.
>
> > It isn't really until page 5 that things really start to take off with novel work.
>
> As above, the paper in general contains many novel contributions. Pages 1-3 are comprised of the introduction, related work, background, and experimental setup. We believe these are relatively tight in their construction, and important for explaining the context behind the results in the paper. In the introduction we also describe Figures 1 (b, c) which is novel and provides an intuition behind our analysis. On page 4 we provide several new demonstrations of the phenomena, for both MLPs and CNNs, including several priors and temperatures, which helps establish the phenomenon and set-up the rest of the paper. We will try to put more focus on the results in later sections in the first pages of the paper in an updated version.
>
> ### Comparison to [1]*
>
> > large parts of the text and figures build heavily off of Izmailov et al. "What are Bayesian Neural Network Posteriors Really Like?".
>
> The only figure that uses the results from [1] is Figure 1 (a), all of the other figures and results are novel and based on original experiments.
>
> > ... it isn't until section 5 that any substantially new results or observations appear to be introduced. The main differences before that point (as far as I can tell) are the addition of MNIST and the use of smaller LeNet-style networks rather than ResNet-20
>
> In Section 4 we also (a) show that the observations in [1] hold for non-Gaussian priors (detailed results in Figure 7); (b) test BNN generalization in domain shift scenario, i.e. CIFAR-10 to STL-10 and MNIST to SVHN (Section 4.2); (c) show that posterior tempering does not resolve the poor generalization of BNNs under covariate shift in contradiction with the hypothesis in [1] (detailed results in Figure 8).
>
>
> ### Comparison to [2] and [3]
>
> Thank you for your comment, we are aware of these works and will add a discussion to the paper. While both our and their work explore failure modes of Bayesian model averaging, we believe that the results and insights are largely orthogonal: in particular, the failure of BMA in [2] and [3] happens fundamentally because the Bayesian model does not contain a reasonable solution in its hypothesis space, which we know is not true in our paper, since the MAP solution is relatively robust to these shifts. Moreover, posterior contraction is an issue in those papers, whereas in our setting the issue is more related to a lack of posterior contraction. Also, as you mention, our setting is around neural networks, out-of-distribution generalization, and new priors, which involve different considerations from these papers.
>
> ### Significance
>
> We believe that the significance of our results is in fact unusually high. We theoretically and empirically shed light on a foundational issue causing a dramatic deterioration in OOD generalization performance for Bayesian neural networks. These results could impact virtually any application of BNNs in the real world, where train and test distributions are often at least slightly different, and we want to make useful predictions. Moreover, based on our analysis we propose a prior which is a first step towards resolving the issue, already showing promising results on a wide range of experiments. We hope that our analysis will provide a foundation to inspire new priors and methods that will further improve the generalization of BNNs under covariate shift.
>
> ### EmpCov helps on MNIST CNN
>
> We apologize, we indeed forgot to include the results for the EmpCov prior on the MNIST CNN, which we will add to the appendix. In fact, *EmpCov* improves performance on all of the noise corruptions. For example, on the Gaussian noise corruption, *EmpCov* achieves 58.3% accuracy while the Gaussian prior achieves 32.7% accuracy; similarly on the impulse noise the results are 96.5% and 90.73% respectively.
>
> However, *EmpCov* does not improve the results significantly on brightness or fog, and even hurts the performance slightly on stripe. In Appendix I, we explain that these corruptions are non-zero mean, and the performance is affected by the sum of the filter weights. We thus propose the *SumFilter* prior which greatly improves the performance on these corruptions.
>
> Overall, both *EmpCov* and *SumFilter* significantly improve the performance of the Bayesian
> CNN on MNIST on a subset of corruptions, covering all the corruptions that we considered.
>
>
> ### Limitations of data-dependent priors
>
> While the *EmpCov* prior depends on the input features, it does not depend on the input labels. As a consequence, *EmpCov* does not suffer from the standard limitations of data-dependent priors such as double-counting the data, which we can see with approaches like empirical Bayes. Indeed, as the Bayesian neural network is only modeling the distribution over $y$ and not over $x$, conditioning the prior on $x$ does not present significant conceptual challenges. We will clarify this point in the text.
>
>
> ### Use of the term covariate shift
>
> When we say that $p(y \vert x)$ is unchanged we mean that there exists a single mapping $g$ which maps inputs $x$ to the corresponding true label distribution, and this mapping is shared between both the train and test data distributions. In particular, if the train data is MNIST and the test data is SVHN, we can assume that such mapping exists as the supports of the two datasets do not overlap. The assumption that a shared $p(y \vert x)$ exists is not important for our analysis, however. We will clarify that our work does not depend on this assumption.
>
>
> ### References
>
> [1] *What Are Bayesian Neural Network Posteriors Really Like?*
> Pavel Izmailov, Sharad Vikram, Matthew D. Hoffman, Andrew Gordon Wilson
>
> [2] *Bayesian averaging of classifiers and the overfitting problem*;
> P Domingos
>
> [3] *Bayesian model averaging is not model combination*;
> TP Minka

---

> > ### Comment · Reviewer_SBkb · 2021-08-16
> > **Edit note**
> >
> > I have increased my score.
> >
> > My main hesitation was around the missing figure for EmpCov, which I can't see because of the revision UI but I'm sure will help.
> >
> > I also like your further observation about MFVI. My score is not conditional on this, but I suggest that you don't just restrict this to the appendix G, but emphasise this in the main body of the paper because it seems important. It also ties into existing work (Farquhar et al. "Liberty or Depth" NeurIPS 2020) that shows MFVI has adequate expressive power with sufficient depth - if it is sufficiently expressive, but also has some beneficial effects under dataset shift, that might encourage us to use MFVI rather than variational approximations that are more expressive in parameter-space. In general, I think it will be interesting for the community to continue to examine the ways in which different approximations offer different trade-offs, which is definitely something your work contributes to an understanding of.

---

> > ### Comment · Reviewer_SBkb · 2021-08-16
> > **Thanks for your response**
> >
> > Thanks for the MC convergence check, that's great. Probably worth doing for non-MNIST things, as that's where I'd expect the biggest trouble with convergence.
> >
> > I appreciate your other responses, especially re Domingos.
> >
> > Not entirely persuaded re 'covariate shift' terminology, but it's not a big deal.

---

### Official Review · Reviewer_rUts · 2021-07-16

**Rating:** 7
**Confidence:** 4

**Summary:**

The paper explores when predictions relying on the posterior predictive distribution of a Bayesian neural network can be poor in the presence of covariate shift and provides a plausible explanation. The main argument being that weakly informative likelihoods cause a lack of posterior contraction, causing the posterior to revert to the prior for at least a subset of the model parameters. The resulting posterior-predictive reverts to the prior predictive and can be detrimental in covariate shift if the shifted data need to exploit this subset of parameters that lack posterior contraction.

**Limitations And Societal Impact:**

The limitations are adequately discussed.

**Main Review:**

I enjoyed reading this paper. It is well written, clearly demonstrates the empirical phenomenon of Bayesian neural networks struggling under certain (but not all) types of covariate shift, and proposes a plausible explanation for the observed phenomenon. Moreover, the fact that the authors used full-batch HMC leads me to believe that the observed issues are likely not a result of approximate inference procedures producing poor approximations and lends credence to the proposed explanation. I only have a few minor quibbles with the paper.

- The paper does a good job of demonstrating that Bayesian model averaging (BMA) is sensitive to perturbations orthogonal to the data subspace when the training data is low rank. What is less clear is whether BMA is as robust as its point-estimated MAP counterpart when the data is full rank or when the perturbations cause shifts in the subspace rather than out of it. I believe Figure 4b was an attempt at showing this. Still, it would be nice to separate it from any idiosyncrasies introduce by the highly curated MNIST dataset and is perhaps best illustrated on simulated data.
- It is interesting that in Figure 4a, for the MLP and CNN (N(0, 0.01) for the high variance (larger PC) components, the posterior is more diffuse than the prior, and there appears to be no posterior contraction. Do the authors have an explanation for this? Is this a case of not seeing enough data (relative to the large number of parameters in the model) or a case of prior misspecification (apriori variances are too small?)?
- Bayesian neural networks are sometimes sold as being effective in the small data regime as a guard against overfitting. However, one would expect the posterior to similarly display a lack of contraction in those cases (there isn’t enough data to condition on). I am curious if the authors believe that the posterior-predictive in such cases would revert to the prior-predictive and result in the BNN ignoring the data?
- Finally, the title of the paper likely needs a qualifier considering that these issues stem from severely over-parameterized models.

**Time Spent Reviewing:**

4.5

---

> ### Author Response · Authors · 2021-08-10
> **Response to Reviewer rUts**
>
> Thank you for your supportive review. We are happy to hear that you enjoyed the paper! Below, we respond to the questions you raised in your review. We hope you can consider these responses, and the broad significance of our contributions, in your final assessment.
>
>
> ### Is BMA insensitive when the corruptions are in the data subspace?
>
> Following your suggestion, we performed an experiment on a synthetic regression problem with full-rank data and verified that the HMC BNNs are similarly robust to Gaussian noise in the inputs as MAP solutions.
>
> We generate synthetic full rank data and train both HMC and MAP (SGD) using a 3-layer synthetic neural network. We then evaluate both methods on a noisy version of the data by adding Gaussian noise. We find that HMC is as robust to noise as MAP for this dataset, which is not the case when we add a column of zeros to the same dataset, where the robustness of HMC is worse than MAP.
> We will provide full details in an updated version of the paper.
>
>
> ### Posterior having higher variance than the prior
>
> We believe that the high marginal variance of the posterior weights is caused primarily by the form of the likelihood. Indeed, suppose the model has 100% train accuracy for some parameters $w$. Then by increasing the weights we can further increase the confidence of the model on all of the training examples.
>
> As an example, suppose our model has ReLU nonlinearities and no bias vectors. Then multiplying the weights in each layer by a constant $c$, the output logits on any input will be multiplied by $c^n$ (see e.g. Proposition 1 in Appendix E of [1]). Consequently, the predicted labels of the model will stay the same on any input, but the confidence (largest predicted softmax probability) will increase, leading to higher value of the likelihood.
>
> As a simple example of the same behaviour, consider the following toy model: $w \in R$ is the parameter with a prior $p(w) = \mathcal N(0, I)$. Suppose the likelihood for a binary observation $y \in \{-1, 1\}$ is given by a sigmoid function using the absolute value of $w$: $p(y \vert w) = \sigma(|w| \cdot y)$. Suppose we only have a single observation $y = 1$. Then the likelihood prefers parameters $w$ with high absolute values, leading to a posterior with higher variance than the prior.
>
>
>
> ### BNNs in small data regime
>
> In fact, we considered the same question when working on the paper, and ran fully-connected BNNs on subsets of the MNIST dataset. Interestingly, we found that the performance of BNNs is indeed inferior to MAP when the training dataset is very small, but the results become more similar as the size of the dataset increases:
>
> |     | 50 datapoints | 100 datapoints | 1000 datapoints |
> |-----|---------------|----------------|-----------------|
> | MAP | 66.4%         | 74.3%          | 90.2%           |
> | BNN | 53.4%         | 65.4%          | 90.3%           |
>
> We believe that the reason why we do not observe the poor generalization of the Bayesian models in the 1000 datapoints regime is that the low-variance directions are fairly consistent across the dataset. However, in extreme low-data cases, we cannot reliably estimate the low-variance directions leading to poor performance according to Propositions 2, 3.
>
>
> ### Title of the paper
>
> While the phenomenon we describe arises when some of the model parameters are unconstrained by the data, we want to clarify that it is relevant to a broad class of models, and does not require the model to be over-parametrized with more parameters than datapoints. For example, any simple neural network with any weights connected to inputs with linear dependencies, such as dead pixels, would be affected by this issue, regardless of how many parameters it has. Several of the models we use are also relatively small. For example, our CNN model on MNIST only has about 62k parameters (for 60k datapoints). We will clarify this point in the updated version of the text.
>
>
>
> ### References
>
> [1] *Bayesian Deep Learning and a Probabilistic Perspective of Generalization;*
> Andrew Gordon Wilson, Pavel Izmailov

---

> > ### Comment · Reviewer_rUts · 2021-08-18
> > **Thank you for your response**
> >
> > The additional experiments and clarifications are helpful. I maintain that this a good paper worthy of publication.

---

### Official Review · Reviewer_SrTJ · 2021-07-16

**Rating:** 7
**Confidence:** 3

**Summary:**

The paper conducts empirical and theoretical analysis on why Bayesian neural networks (BNNs) with high-fidelity inference schemes (HMC) are vulnerable to out-of-distribution data. The authors empirically noticed that BNNs perform worse than the point estimation MAP on the corrupted test dataset. They propose that it is the linear dependency in the input features that causes this lack of robustness. This input linear dependency will not provide likelihoods for the weight distributions in the orthogonal direction, thus won't modify the weights' prior distributions. Those unmodified weight distributions are vulnerable to input perturbations. Based on these observations, the authors propose a new prior covariance to providing more robustness.

**Limitations And Societal Impact:**

The mentioned limitations are good. Practitioners are also interested to see the results or mentions of another approximate inference scheme -- variational inference -- which is more computationally efficient and widely used.

**Main Review:**

The paper is well-written, and the delivery is clear. The theoretical perspective is interesting. But the empirical demonstration is not that novel because a similar observation exists as the author mentioned, although the paper provides more evidence. Furthermore, the paper would be more motivated if the following two points can be addressed:
- In the first place, the analyses are all interesting, but why would people want to apply their model to out-of-distribution data? Aren’t people expected to detect the ODDs first and exclude them from the testing dataset?
- In Fig. 4, it seems that the prior already has a quite small scale. So sampling from those priors will generate weights almost zero (if my understanding is correct), which is somewhat the opposite of the paper's argument. Will this small error during testing propagate through the network? (Please correct me if my interpretation is wrong.)

**Time Spent Reviewing:**

3.5

---

> ### Author Response · Authors · 2021-08-10
> **Response to Reviewer SrTJ**
>
> Thank you for your review and questions! Below we address your questions and hope that you will consider raising your score in your final evaluation. We will add clarifications to the updated version of the paper based on the points you raised.
>
> ### Novel contributions.
>
> We want to emphasize that while the observation that BNNs generalize poorly on OOD data was briefly observed in [1], our work has a number of important novel contributions. First, we show that this phenomenon holds generally for a variety of parameter-space priors, new model architectures and domain shift corruptions (Section 4.2). Furthermore, [1] hypothesized that posterior tempering would address the poor OOD generalization of BNNs, which we show is not the case. Then, we provide a detailed analysis, both theoretically and empirically, showing that linear dependencies in the data play a role in the poor robustness of BNNs. Based on our new understanding, we additionally propose a new prior, and validate that this prior significantly improves the robustness of BNNs to covariate shift.
>
> ### Should we just detect OOD and not make predictions?
>
> In most real-world applications of deep learning we are very likely to encounter some degree of covariate shift, where the test data distribution differs from the train data [2]. It is thus important that our models are able to provide meaningful predictions under covariate shift, and not simply detect it. Indeed, if our model refuses to make predictions on any of the test datapoints, labeling them as OOD, the model is no longer useful.
>
> In all the settings considered in this paper, the test data is still clearly recognizable. See for example [2] for the description of CIFAR-10-C corruptions. Indeed, being able to make useful predictions under noise corruptions and mild domain shift, where the data are still clearly recognizable, is fundamental to the applicability of Bayesian neural networks in the real-world. In such settings, a good representation of uncertainty should not simply say “I don’t know”.
>
> Finally, we note that multiple papers have evaluated generalization of both standard and approximate Bayesian neural networks to covariate shift in the same settings that we consider, for example [2, 3, 4, 5] and others.
>
>
>
> ### The prior is already small-variance
>
> While the prior variance is indeed on the order of $\frac 1 {100}$ for many of the experiments we conducted in the paper, the noise in the samples still has a large effect on the outputs of the model. The prior determines the magnitude of all the weights in the first layer, both the parameters that are constrained by the data, and the parameters for which the posterior coincides with the prior (Figure 4, a). As a result, the noisy unconstrained parameters, while being small in absolute value, will have a large contribution on the predictions of the model.
>
> As an example, consider a fully-connected first layer with ReLU nonlinearity in a neural network model (the derivation is analogous for convolutional models):
> $z =  \max(W x + b, 0)$,
> where $W$ are the weights and $b$ are the biases of the first layer. If we multiply the parameters in the first layer by a positive constant $c$, the outputs of the layer would also simply be scaled by $c$: $ \max(c \cdot W x + c \cdot b, 0) = c \cdot \max(W x + b, 0) = cz$. So, the scale of the parameters in the first layer affect the scale of its outputs, but the signal-to-noise ratio in the outputs of the first layer does not depend on this scale.
>
> Finally, we note that in Figure 7 (a), we considered higher variance priors, sometimes with even more poor robustness.
>
>
> ### Variational Inference
>
> Please see our common message to all reviewers for a detailed response, where we provide new theoretical and empirical insights into the behavior of variational inference under covariate shift. Intuitively, the commonly used mean-field variational inference (MFVI) lacks the capacity to learn the varaitional posterior over unconstrained directions in the parameter space that would match the prior. Consequently, MFVI is not as affected by the issues identified in Propositions 2, 3.
>
>
> ## References
>
> [1] *What Are Bayesian Neural Network Posteriors Really Like?*
> Pavel Izmailov, Sharad Vikram, Matthew D. Hoffman, Andrew Gordon Wilson
>
> [2] *Benchmarking Neural Network Robustness to Common Corruptions and Perturbations*;
> Dan Hendrycks, Thomas Dietterich
>
> [3] *Can You Trust Your Model's Uncertainty? Evaluating Predictive Uncertainty Under Dataset Shift*; Yaniv Ovadia, Emily Fertig, Jie Ren, Zachary Nado, D Sculley, Sebastian Nowozin, Joshua V. Dillon, Balaji Lakshminarayanan, Jasper Snoek
>
> [4] *Bayesian Deep Learning and a Probabilistic Perspective of Generalization*; Andrew Gordon Wilson, Pavel Izmailov
>
> [5] *Efficient and Scalable Bayesian Neural Nets with Rank-1 Factors*;
> Michael W. Dusenberry, Ghassen Jerfel, Yeming Wen, Yi-An Ma, Jasper Snoek, Katherine Heller, Balaji Lakshminarayanan, Dustin Tran

---

> > ### Comment · Reviewer_SrTJ · 2021-08-16
> > **Thanks for the response**
> >
> > Thanks for the explanation and clarification. I will increase my score to 7 for acceptance.
> >
> > My main hesitation was the validity of the application to OOD data. The OOD-related area is complex and already involves uncertainty quantification, OOD detection. It is worth knowing that applications to datasets with small perturbations are a solid topic. Interesting work!

---

### Official Review · Reviewer_8WYY · 2021-07-19

**Rating:** 7
**Confidence:** 3

**Summary:**

This paper shows that Bayesian neural networks are vulnerable to distribution shifts. The authors provide theoretical results showing that this vulnerability is caused by the linear dependencies in the training samples, which are supported by their experiments as well. Based on this theoretical insight, the authors propose a simple initialization method to resolve this undesirability.


**Limitations And Societal Impact:**

The authors properly addressed the limitation in the paper. Also, the authors provided the potential negative social impact caused by the intensive computational load of HMC.

**Main Review:**

Evaluating and improving the robustness of Bayesian neural networks to distribution shifts is an important research area getting increasing attention in the community. In this regard, this paper provides valuable insights that explain why Bayesian neural networks are vulnerable to covariate shifts and how to overcome this vulnerability.

In addition, the paper is well-written, intuitive, and easy to follow. I think that the motivating examples in Fig. 1 and Chapter 4 are informative to both experts and readers who are not familiar with the Bayesian neural networks’ vulnerability to the distribution shift. Also, the structure of Chapter 5, starting from the simplest case in which all training samples have a constant zero at a particular feature with feed-forward neural nets and then generalizing the case, helps readers get insights into the phenomenon. Also, the theoretical result in Proposition 3 covers sufficiently large cases with mild assumptions. Finally, the proposed initialization method is simple and effective.

I believe this is a good paper, and I have only two minor comments to authors:

1. Regarding Fig. 1 (b), the authors present BNN weights sampled HMC and claim that they are unstructured. However, the features obtained by marginalization might have the structured form, e.g., zeros, or small values, on the boundaries. Considering the posterior weights on the boundary location would follow zero-mean Gaussian (or Laplace depending on the prior), I think this case is highly likely. If so, I’m wondering if the authors can still argue that the BNN weights are unstructured.

2. Looking at the theoretical results in Chapter 5, there are no restrictions on the way to perform posterior inference of BNNs. However, I’m wondering why the authors did not provide results for BNNs with stochastic VI, even though they are more frequently used in practice.


**Time Spent Reviewing:**

15

---

> ### Author Response · Authors · 2021-08-10
> **Response to Reviewer 8WYY**
>
> Thank you for your supportive review! Below, we address your questions in detail. In particular, for variational inference, we provide new theoretical and empirical insights inspired by your questions (see our post "Response to reviewers"). We hope that you will consider these results, as well as the timeliness and significance of this work, in your final evaluation.
>
> **Are the weights in Figure 1 (b) unstructured?**
> In Figure 1 (b) we show two samples from the posterior which are both very noisy compared to a structured MAP solution. Indeed, if we computed the expected weights of the first layer under the posterior, we would observe more structure: the weights near the boundaries as well as projections of the weights on the unconstrained directions will be zero in the expectation. Note however that in Bayesian model averaging we compute the integral of the predictions, e.g., averaging the functions, but we do not average the weights themselves:
> $p(y \vert x) = \int p(y \vert x, w) p(w \vert D) dw \ne p(y \vert x, \mathbb E_{p(w \vert D)} w)$. In other words, due to the nonlinearity of neural networks, the randomness in the unconstrained weights does not simply average out to zero, and instead has a non-trivial effect on the predictions.
>
>
> ### Variational inference
>
> Please see our common message to all reviewers for a detailed response. We provide both theoretical and empirical insights into the behavior of VI under covariate shift. Mean Field VI is constrained in the types of variational distributions it can use, and it cannot represent the posterior correctly in the unconstrained directions in the parameter space; consequently, it does not suffer from the issues identified in our work as much as precise HMC inference.

---

### Author Response · Authors · 2021-08-10
**Response to reviewers**

We would like to thank all the reviewers for thoughtful feedback, and supportive comments. We wish to emphasize that the paper is making several substantial contributions: (1) we demonstrate in a variety of settings that both fully-connected and convolutional Bayesian neural networks have a dramatic deterioration of accuracy in covariate shift settings; (2) we provide a new theoretical and empirical understanding of this result; (3) directly based on this new understanding, we propose a prior to help resolve these issues; (4) we vigorously demonstrate that the proposed *EmpCov* prior provides both good generalization accuracy and log likelihoods over a wide variety of shifts. It is quite rare to have all of these types of contributions in one paper, and the fact that we can propose a relatively successful new approach based on our analysis lends strong support to the relevance of the analysis. We also note that while conceptually oriented papers can be hard to evaluate, they are really important to progress in the field.

Moreover, we believe these contributions are timely and have unusually broad significance. Indeed, the understanding provided in contribution (2) could affect virtually every real-world application of Bayesian neural networks, where we often have training and test points from at least slightly different distributions, but we still want to make useful predictions. The *EmpCov* prior is an important step in this direction, and the understanding provided into (2) provides a mechanism for a great deal of future methodological innovation.

Inspired by reviewer comments, we have conducted several novel experiments during the rebuttal phase:
- We evaluated the robustness of variational inference (VI) on the MLP on MNIST. We also provide a detailed theoretical analysis supporting this experiment (reviewers 8WYY, SrTJ).
- We ran an experiment evaluating the robustness of BNN and MAP solutions on full-rank data in a synthetic regression problem (reviewer rUts).
- We compared the in-distribution performance of the HMC and MAP solutions in the low-data regime (reviewer rUts).
- We estimated the convergence of the MC integration as a function of the number of samples (reviewer SBkb).

In our response to reviewer SBkb we also provide the results for the *EmpCov* prior using CNN on MNIST, which we forgot to include in the supplementary.

We provide individual responses to reviewers as separate posts, including these new results inspired by reviewer comments. Below, we provide insights into variational inference, which was a common question raised by reviewers 8WYY, SrTJ.


### Comments on Variational inference

Inspired by reviewer comments, we have performed an analysis to theoretically characterize the behaviour of VI under covariate shift and evaluated robustness of VI empirically, as follows:


#### Theoretical analysis

Suppose the prior is $p(w) = \mathcal N(0, \alpha^2 I)$ and the variational family contains distributions of the form $q(w) = \mathcal N(\mu, \Lambda)$, where the mean $\mu$ and the covariance matrix $\Lambda$ are parameters. Variational inference solves the following optimization problem:
maximize $\mathbb E_{w \sim \mathcal N(\mu, \Lambda)} p(D \vert w) -
KL\left(\mathcal N(\mu, \Lambda) \vert\vert \mathcal N(0, \alpha^2 I)\right)
$ wrt $\mu$, $\Lambda$.

First, let us consider the case when the parameters $\Lambda$ are unconstrained (can be any positive-definite matrix). Suppose we are using a fully-connected network, and there exists a linear dependence in the features, as in Proposition 2. Then, there exists a direction $d$ in the parameter space of the first layer of the model, such that the projection of the weights on this direction will not affect the likelihood, and the posterior over this projection will coincide with the prior and will be independent from other directions (Proposition 2), which is Gaussian. Consequently, the optimal variational distribution will match the prior in this projection, and will also be independent from the other directions (in other words, $d$ will be an eigenvector of the optimal $\Lambda$ with eigenvalue $\alpha^2$). So, variational inference with a general Gaussian variational family will suffer from the same exact issue that we identified for the true posterior. Furthermore, we can generalize this result to convolutional layers completely analogously to Proposition 3.

Now, let us consider the mean-field variational inference (MFVI) which is commonly used in practice in Bayesian deep learning. In MFVI, the covariance matrix $\Lambda$ is constrained to be diagonal. Consequently, for general linear dependencies in the features the variational distribution will not have sufficient capacity to make the posterior over the direction $d$ independent from the other directions. As a result, MFVI will not suffer as much as exact Bayesian inference from the issue presented in Propositions 2, 3.

One exception is the dead pixel scenario described in Section 5.1, where one of the features in the input is a constant zero. In this scenario, MFVI will have capacity to make the variational posterior over the corresponding weight match the prior, leading to the same lack of robustness described in Proposition 1.

We will add a detailed discussion to appendix G.

**Note on optimization.** The theoretical analysis above assumes that we are able to solve the VI optimization problem perfectly. In practice, the robustness of VI might depend on the choice of optimizer and hyper-parameters, but we would expect the same qualitative behavior.

#### Empirical support

In addition to the empirical analysis above, we ran mean field variational inference on our fully-connected network on MNIST and evaluated robustness on the MNIST-C corruptions. Below we report the accuracy for MFVI, MAP and HMC BNN with a Gaussian prior:

| method   |   MNIST |   gaussian_noise |   motion_blur |   scale |   brightness |   stripe |   canny_edges |
|:---------|-----------:|-----------------:|--------------:|--------:|-------------:|---------:|--------------:|
| MAP      |     0.985  |            0.706 |        0.8666 |  0.7756 |       0.5063 |   0.341  |        0.6798 |
| MFVI     |     0.9794 |            0.625 |        0.8216 |  0.7054 |       0.6883 |   0.4765 |        0.6993 |
| HMC     |     0.9819 |            0.432 |        0.8214 |  0.6945 |       0.2231 |   0.348 |        0.6334 |

As expected from our theoretical analysis, MFVI is much more robust to noise than HMC BNNs. However, on some of the corruptions (*gaussian_noise*, *motion_blur*, *scale*) MFVI underperforms the MAP solution. At the same time, MFVI even outperforms MAP on *brightness*, *stripe* and *canny_edges*.

---

### Decision · Program_Chairs · 2021-09-28

**Decision:**

Accept (Poster)

**Comment:**

The paper explores when predictions relying on the posterior predictive distribution of a Bayesian neural network can be poor in the presence of covariate shift and provides a plausible explanation. Weakly informative likelihoods cause a lack of posterior contraction, causing the posterior to revert to the prior for a subset of the model parameters. The resulting posterior-predictive reverts to the prior predictive and can be detrimental in covariate shift.

The paper convincingly demonstrates the empirical phenomenon of Bayesian neural networks struggling under certain types of covariate shifts and proposes a plausible explanation for the observed phenomenon. It is solid in terms of its empirical analysis. Furthermore, it presents a new prior that does not suffer from the downsides of the conventional priors commonly used.

Initial doubts about a lack of samples to evaluate the posterior predictive distribution were successfully addressed in the discussion period. Likewise, initial concerns about a lack of novelty with respect to [Izmailov et al., ICML 2021] were also clarified. Another concern was that the proposed phenomenon might not be the only explanation for BNNs failing in OOD situations, but the reviewers didn't consider this a significant weakness of the paper.

**Consistency Experiment:**

NeurIPS has a long history of experimentation. In 2014, NeurIPS ran an experiment in which 10% of submissions were reviewed by two independent committees to quantify the randomness in the review process. This year, we repeated a variant of this experiment to see how the quality of the review process has changed over time.  This paper was part of the experiment and was therefore assigned to two committees (consisting of reviewers, an Area Chair, and a Senior Area Chair) that reached independent decisions.  If both committees made the same recommendation, this recommendation was followed. If a single committee recommended acceptance, the paper was accepted (with the exception of a few cases in which the other committee identified what we considered a fatal flaw, e.g., an error in a key result).

Both committees reached the same decision: **Accept (Poster)**

The other committee assigned to the paper recommended **Accept (Poster)**.  You can find the other set of reviews, along with any follow up discussion with the authors here:
https://openreview.net/forum?id=T1r6y8PnVGk